

# The neXtSIM-DG dynamical core: A Framework for Higher-order Finite Element Sea Ice Modeling

Thomas Richter[1], Véronique Dansereau[2], Christian Lessig[3], and Piotr Minakowski[1]

[1]Otto-von-Guericke University Magdeburg, Germany. Institute for Analysis and Numerics. numerics.ovgu.de
[2]Institut des Sciences de la Terre, Université Grenoble Alpes, CNRS (UMR5275), Gières, France
[3]Otto-von-Guericke University Magdeburg, Germany. Institute for Simulation and Graphics

**Correspondence:** Thomas Richter (thomas.richter@ovgu.de)

**Abstract.** The ability of numerical sea ice models to reproduce localized deformation features associated with fracture processes is key for an accurate representation of the ice dynamics and of dynamically coupled physical processes in the Arctic and Antarctic. Equally key is the capacity of these models to minimize the numerical diffusion stemming from the advection of these features, to ensure that the associated strong gradients persist in time, without the need to unphysically re-inject energy for re-localization. To control diffusion and improve the approximation quality, we present a new numerical core for the dynamics of sea ice that is based on higher order finite element discretizations for the momentum equation and higher order discontinuous Galerkin methods for the advection. The mathematical properties of this core are discussed and detailed description of an efficient shared memory parallel implementation is given. In addition, we present different numerical tests and apply the new framework to a benchmark problem to quantify the advantages of the higher order discretization. These tests are based on Hibler's viscous-plastic sea ice model, but the implementation of the developed framework in the context of other physical models reproducing a strong localization of the deformation are possible.

## 1 Introduction

Sea ice plays a critical role for the development of the Earth system with up to $15\%$ of the world's oceans being covered by it at some point during the year. It is, for example, an important part of the global energy budget and its high albedo keeps arctic oceans cool, affecting global oceanic circulations. An accurate simulation of sea ice is therefore of importance, in particular to describe the evolution and impact of climate change. The numerical modeling of sea ice is, however, very challenging since it is characterized by nonlinear and highly localized processes.

In the present work, we develop a numerical scheme for sea ice that use higher order finite elements for the sea ice momentum and the advection equations and specifically aims to provide a high fidelity discretization with small numerical diffusion and good approximation properties. We choose discontinuous Galerkin methods for the advection because they allow a Eulerian treatment of the equations of motion that is compatible with the habits of the sea ice and climate modelling community while extending naturally to high order and exhibiting limited numerical diffusion. The momentum equation will be formulated in a variational finite element way that also allows naturally for higher order schemes and allows for a direct coupling to the





discontinuous Galerkin advection discretization. The proposed numerical scheme will form the dynamical core in the next

neXtSIM-DG sea ice model that is currently under development.

Since the first extended in-situ observational campaigns of the 1970's in the Arctic, sea ice has been identified as a densely fractured material in which most of the deformation is taking place locally by "relative motion at the cracks" with the ice between the cracks being virtually "rigid" (Coon et al., 1974). This relative motion of ice plates, referred to in the sea ice community as *floes*, translates into three main deformation processes: opening of fractures; joining along larger features called

*leads*; the shearing along opened fractures and the closing of leads, resulting in the formation of pressure ridges. Although highly localized around cracks, the processes play a key role in the polar ocean systems by governing the location and intensity of bio-chemical processes and the exchange of heat, mass and momentum between the ice, ocean, and atmosphere, e.g. (Marcq and Weiss, 2012; Vihma, 2014; Goosse et al., 2018; Horvat and Tziperman, 2018; Taylor et al., 2018). Importantly, the three processes also determine to a significant extent the large-scale mechanical resistance of the ice cover and hence its mobility

and the overall rates of ice export out of the Arctic (e.g., Rampal et al., 2009, 2011).

Satellite remote sensing data, such as the RADARSAT Geophysical Processor System sea ice motion products which became available in the late 1990's, have allowed for the observation of these localized processes at the global scale of the Arctic Ocean. The term "Linear Kinematic Features" (LKFs) was then proposed to designate the associated near-linear zones of discontinuities in the drift velocity fields. These LKFs correspond to areas with a high density of fractures in the ice cover,

which strongly concentrates its deformation (Kwok, 2001). In recent years, a large number of observational analyses of sea ice deformation data, e.g. Lindsay and Stern (2003); Marsan et al. (2004); Rampal et al. (2008); Stern and Lindsay (2009); Hutchings et al. (2011); Oikkonen et al. (2017), has fuelled a race in the modelling community towards a better reproduction of LKFs in thermodynamical models, in particular, with respect to their spatial and temporal statistics, e.g. (Girard et al., 2011; Rampal et al., 2016; Hutter et al., 2018; Rampal et al., 2019; Bouchat et al., 2022). Different approaches have been

taken towards this goal : new mechanical (i.e., rheological) continuum models have been proposed for sea ice (Schreyer et al., 2006; Sulsky and Peterson, 2011; Girard et al., 2011; Dansereau et al., 2016; Ólason et al., 2022), the mechanical parameters of existing models have been tuned (Bouchat and Tremblay, 2017), and the spatial resolution of models has been increased (Hutter et al., 2018).

The ability to reproduce adequately LKFs in continuum sea ice models however raises an equally important challenge: that

of keeping the very strong gradients in sea ice properties (e.g. velocity, thickness, concentration) that stem from the extreme localization of the deformation as the ice is advected by winds and ocean currents. This numerical discretization problem is, in fact, not unique to sea ice but encountered for all materials that are experiencing both highly localized deformations resulting from brittle fracturing processes and high post-fracture strains. Another important example from the geosciences is the Earth crust, where brittle processes leading to strain localization and slip coexist in faults, landslides and volcanic edifices, e.g. (Peng

and Gomberg, 2010; Burov, 2011). Sea ice, however, represents an extreme case as it is constantly moving and experiencing much larger relative deformations and drift velocities (about $7\,\mathrm{cm/s}$ on average).

Several numerical approaches have been studied and dedicated advection schemes have been developed to limit numerical diffusion in models of the Earth crust, (e.g., see Zhong et al. (2015) for a review). In the sea ice modelling community however,





the treatment and, in particular, the quantification of numerical diffusion of advected gradients has received relatively little
attention. Notable exceptions are the works by Flato (1993) and Huang and Savage (1998), which applied particle-in-cell
methods to treat the advection of strong gradients in ice concentration and thickness, not associated with LKFs but with the
migration of the edge of the Arctic sea ice cover (the so-called "ice edge"), Lipscomb and Hunke (2005), which used an
incremental remapping to limit diffusion, Sulsky and Peterson (2011) which introduced the Material Point Method and tested
its robustness in the context of sea ice by performing idealized convection benchmark problems and Danilov et al. (2015),
which employed a flux corrected Taylor-Galerkin method. NeXtSIM (Rampal et al., 2016) is based on a Lagrangian model
and hence completely avoids diffusion during transport, although remeshing operations are required in this framework which
themselves induce some diffusion. The implementation of discontinuous Galerkin methods to treat the advection of sea ice was
first proposed by Dansereau et al. (2016, 2017) and used with higher orders, with a quantification of diffusion, by Dansereau
et al. (2021). Mehlmann et al. (2021b) compared sea ice simulations using different meshes, mesh resolutions and advection
schemes. However, the focus of their paper was the discretization of the momentum equation and no specific discussion of
numerical diffusion was given.

   **Outline.** The following section will introduce the basic equations and the notation used throughout the manuscript. We limit
ourselves to the most widely used dynamical framework, which is the so-called Visco-Plastic rheology (Hibler, 1979), to focus
on the discretization and to aid comparison to other numerical schemes in the literature. We will extend the discretization to
more recently developed "Elasto-Brittle" schemes (MEB and BBM, e.g. Dansereau et al. (2016); Ólason et al. (2022)) else-
where. The third section details the numerical discretization of the sea ice model, including the advection and the momentum
equations. Section 4 focuses on the implementation as well as on the shared-memory parallelization of the numerical model.
Finally, in Section 5 we consider basic tests to validate the method and apply it to established benchmark problems (Mehlmann
and Korn, 2021). The paper concludes with an outlook.

## 2   Governing equations

We denote by $\Omega \subset \mathbb{R}^2$ the two-dimensional domain of the sea ice. The sea ice models we investigate consist of a momentum
equation for the velocity field $\mathbf{v} : \Omega \to \mathbb{R}^2$ and further advection equations for tracer variables. In simple models, such as the
one introduced by Hibler (1979), the tracers are usually the ice height $H : \Omega \to [0, \infty) \subset \mathbb{R}$ and ice concentration $A : \Omega \to$
$[0, 1] \subset \mathbb{R}$. Here, we consider the following system of sea ice equations

$$\rho_{\text{ice}} H \partial_t \mathbf{v} = \text{div } \boldsymbol{\sigma} + A \boldsymbol{\tau}(\mathbf{v}) - \rho_{\text{ice}} H f_c \boldsymbol{e}_z \times \mathbf{v} - \rho_{\text{ice}} H g \nabla \tilde{H}_g,$$
85                                                                                                                    (1)
$$\partial_t A + \text{div}(\mathbf{v}A) = 0, \quad \partial_t H + \text{div}(\mathbf{v}H) = 0.$$

Here, $\rho_{\text{ice}}$ is the ice density, $f_c \boldsymbol{e}_z \times \mathbf{v}$ is the Coriolis term with Coriolis parameter $f_c$ and vertical unit vector $\boldsymbol{e}_z$, $g$ is the
gravitational acceleration, and $\tilde{H}_g$ the sea surface height. We focus on a stand alone dynamics model without coupling to an
ocean and an atmospheric model. Following Coon (1980), we approximate the surface height by the Coriolis term

$$-\rho_{\text{ice}} H g \nabla \tilde{H}_g \approx \rho_{\text{ice}} H f_c \boldsymbol{e}_z \times \mathbf{v}_o,$$



where $\mathbf{v}_o$ is the ocean surface velocity. The forcing $\boldsymbol{\tau}(\mathbf{v})$ is given by

$$\boldsymbol{\tau}(\mathbf{v}) = C_o \rho_o \|\mathbf{v}_o - \mathbf{v}\|_2 \cdot (\mathbf{v}_o - \mathbf{v}) + C_a \rho_a \|\mathbf{v}_a\|_2 \cdot \mathbf{v}_a.$$

The index "$o$" represents the *ocean* with the surface drag $C_o$, the water density $\rho_o$, and again the ocean surface velocity $\mathbf{v}_o$ while "$a$" denotes the *atmosphere* with drag coefficient $C_a$, density $\rho_a$ and wind field $\mathbf{v}_a$. We neglect turning angles and thermodynamic effects in Eq. (1). Therefore, the constraints $A \in [0,1]$ and $H \in [0,\infty)$ are not naturally enforced by the equations but must be ensured by projections. In the following, we will use the following notation of the momentum equation (with the approximation of the surface height)

$$\rho_{\text{ice}} H \partial_t \mathbf{v} = \operatorname{div} \boldsymbol{\sigma} + F(\mathbf{v}), \quad F(\mathbf{v}) = A\boldsymbol{\tau}(\mathbf{v}) + \rho_{\text{ice}} H f_c \mathbf{e}_z \times (\mathbf{v}_o - \mathbf{v}). \tag{2}$$

Model (1) is closed by specifying a rheology, i.e., the relation between the (vertically integrated) stress $\boldsymbol{\sigma}$ and the strain rate $\epsilon$,

$$\boldsymbol{\epsilon}(\mathbf{v}) = \frac{1}{2}(\nabla \mathbf{v} + \nabla \mathbf{v}^T), \quad \boldsymbol{\epsilon}'(\mathbf{v}) = \boldsymbol{\epsilon}(\mathbf{v}) - \frac{1}{2}\operatorname{tr}\left(\boldsymbol{\epsilon}(\mathbf{v})\right)I,$$

as well as the ice tracer quantities $H$ and $A$ (and possibly further parameters). Different rheological models have been proposed in the literature. As this paper focuses on computational questions that are largely independent of the chosen rheology, we consider the most widely used one, i.e. the *viscous-plastic* (VP) model proposed by Hibler (1979). It prescribes

$$\sigma(\mathbf{v}) = 2\eta \boldsymbol{\epsilon}'(\mathbf{v}) + \zeta \operatorname{div}(\mathbf{v})I - \frac{P}{2}I, \tag{3}$$

with viscosities $\eta, \zeta$ that, using the notation introduced in Mehlmann and Richter (2017), are given by

$$\eta = \frac{\zeta}{e^2}, \quad \zeta = \frac{P_0}{2\sqrt{\Delta_{\min}^2 + \operatorname{tr}(\epsilon)^2 + 2e^{-2} \cdot \boldsymbol{\epsilon}' : \boldsymbol{\epsilon}'}}. \tag{4}$$

Here $e = 2$ is the excentricity of the elliptical yield curve, $\Delta_{\min} > 0$ is the threshold defining the transition to a viscous behaviour for very small strain, $P_0$ is the ice strength, and $P$ is the replacement pressure

$$P_0 = P^\star \cdot H \cdot \exp\left(-C(1-A)\right), \quad P = \frac{\Delta(\epsilon)}{\Delta_{\min} + \Delta(\epsilon)} \cdot P_0. \tag{5}$$

Common default values for the model parameters $\rho_{\text{ice}}, \rho_a, \rho_w, e, C, P^\star$ can be found in Tab. 1.

The VP model is highly nonlinear. Therefore, a solution with implicit methods is very challenging, see Losch et al. (2014); Mehlmann and Richter (2017); Shih et al. (2022) for various approaches based on Newton's method. Picard iterations are also slow and an explicit time-stepping would require excessively small time steps (Ip et al., 1991). Hence, the so-called Elastic-Visco-Plastic (EVP) model is a widely used variant of the VP rheology (Hunke, 2001; Kimmritz et al., 2016). It adds a pseudo-elastic behaviour to improve numerical performance. The constitutive law (3) is in this case given by

$$\frac{1}{E} \cdot \frac{d}{dt}\boldsymbol{\sigma} + \boldsymbol{\sigma} = \boldsymbol{\sigma}(\mathbf{v}), \tag{6}$$

where $\sigma(b\mathbf{v})$ is the VP-relation given by Eq. (3). EVP should, however, be considered as a model different from VP since its solutions do not converge to the VP ones. An alternative variant that can be considered as a pseudo-time-stepping scheme is the mEVP scheme (Bouillon et al., 2013), see Section 3.4. The mEVP scheme converges to the VP solution given a sufficiently large number of iterations.





## 3 Higher order finite element discretization of the sea ice equations

In the following, we describe the discretization of the sea ice equations (1) in space and time using higher-order finite elements. All tracers and also the strain rate tensor $\epsilon$ and the stresses $\sigma$ are discretized with a discontinuous Galerkin (dG) approach whereas the ice velocity $\mathbf{v}$ is discretized using quadratic continuous finite elements.

### 125 3.1 Mesh domain

Discretizations of the sea ice equations are typically used within a coupled Earth system model. One consequence is that the time step of the numerical sea ice model is not only determined by the desired accuracy and stability considerations but also constrained by the atmospheric and oceanic components of the Earth system model.

By $\Delta t$ we denote the time step of the sea ice equations. In coupled applications, $\Delta t$ is usually the time step $\Delta t_o$ of the ocean model or a multiple of it, i.e. $\Delta t = \Delta t_o/k$, for $k \in \mathbb{N}$. Although dynamic time discretizations with varying step sizes are possible, we will only consider uniform time steps with $\Delta t_n = \Delta t$ for all steps $n$. The time mesh is hence given by

$$t_0 < t_1 < t_2 < \cdots < t_N = T, \quad \Delta t := t_n - t_{n-1}. \tag{7}$$

Assuming a typical ocean model, we will have $\Delta t$ in the range $\Delta t \in [30\,\mathrm{s}, 240\,\mathrm{s}]$. Given typical ice velocities $|\mathbf{v}|_\infty \leq 1\,\mathrm{m}\cdot\mathrm{s}^{-1}$, explicit time-stepping will be stable for mesh sizes up to a resolution of

$$\Delta x \approx C_r \cdot \|\mathbf{v}\| \cdot \Delta t \approx C_r \cdot 250\,\mathrm{m}, \tag{8}$$

where $C_r$ is a constant that depends on the degree $r$ of the time stepping scheme. The factor $C_r$ scales like $C_r \approx 2r + 1$ (Chalmers and Krivodonova, 2020). Hence, for a dG(2) method and time stepping scheme of balanced order with $r = 2$, the minimum mesh element size should be larger than $2\,\mathrm{km}$ if a time step of $\Delta t = 240\,\mathrm{s}$ is used.

For the spatial discretization, we employ a parametric finite element mesh of the domain $\Omega$. To be consistent with the sea ice and climate modelling communities, we base the discretization on quadrilateral meshes $\mathcal{T}_h$ (as opposed to triangular ones). The meshes are topologically fully regular but geometrically unstructured and consist of nodes $\mathbf{x}_{i,j}$, elements $T_{i,j}$ and edges $e_{i,j}^{\{x,y\}}$, such that

$$\mathcal{T}_h = \begin{cases} \mathbf{x}_{i,j} \in \Omega & i = 0, \ldots, N_x, \ j = 0, \ldots, N_y, \\ T_{i,j} = (\mathbf{x}_{i-1,j-1}, \mathbf{x}_{i,j-1}, \mathbf{x}_{i-1,j}, \mathbf{x}_{i,j}) & i = 1, \ldots, N_x, \ j = 1, \ldots, N_y, \\ e_{i,j}^x = (\mathbf{x}_{i-1,j}, \mathbf{x}_{i,j}) & i = 1, \ldots, N_x, \ j = 0, \ldots, N_y, \\ e_{i,j}^y = (\mathbf{x}_{i,j-1}, \mathbf{x}_{i,j}) & i = 0, \ldots, N_x, \ j = 1, \ldots, N_y \end{cases}, \tag{9}$$

where $N_x, N_y \in \mathbb{N}$ denote the number of elements in $x$- and $y$-direction. See Fig. 1 for a depiction. The nodes are lexicographically ordered, i.e. $k = i + (N_x + 1)j$ is the consecutive index. Each geometric mesh element $T_{i,j}$ can be defined via a mapping

$$\mathbf{T}_{i,j} : \hat{T} := (0,1)^2 \mapsto T_{i,j}$$



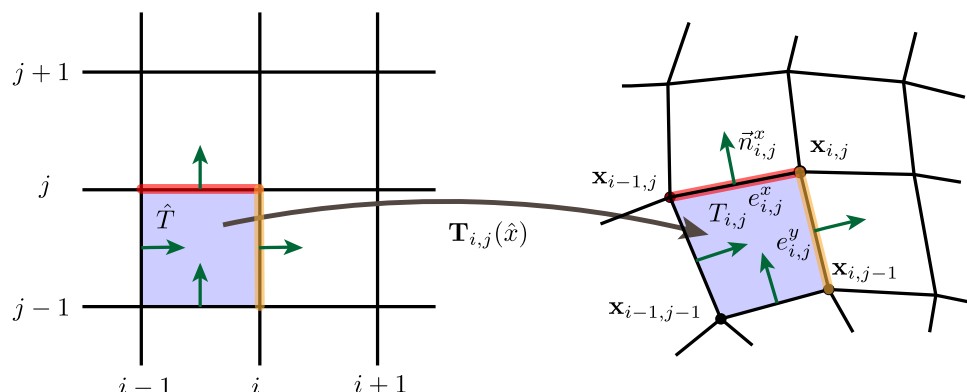

**Figure 1.** Parametric mesh. Each element $T_{i,j} \in \mathcal{T}_h$ (on the right) arises from the a mapping $\mathbf{T}_{i,j} : \hat{T} \to T_{i,j}$ from the reference element $\hat{T} = (0,1)^2$ (on the left). The mesh elements $T_{i,j}$ are general quadrilaterals such that the mappings $\mathbf{T}$ are bi-linear polynomials. The edges $e_{i,j}^x$ and $e_{i,j}^y$ are straight lines.

from a unique reference element $\hat{T}$ using the bi-linear polynomial

$$\mathbf{T}_{i,j}(\hat{\mathbf{x}}) = (1-\hat{\mathbf{x}}_1)(1-\hat{\mathbf{x}}_2)\mathbf{x}_{i-1,j-1} + \hat{\mathbf{x}}_1(1-\hat{\mathbf{x}}_2)\mathbf{x}_{i,j-1} + (1-\hat{\mathbf{x}}_1)\hat{\mathbf{x}}_2\mathbf{x}_{i-1,j} + \hat{\mathbf{x}}_1\hat{\mathbf{x}}_2\mathbf{x}_{i,j}, \tag{10}$$

see again Fig. 1. On each edge $e_{i,j}^x$ and $e_{i,j}^y$, we consider one unit normal vector. Its orientation arises from mapping the unit normal vectors $\hat{e}^x = (1,0)^T$ and $\hat{e}^y = (0,1)^T$ of the reference element to the edges of $\mathcal{T}_h$.

### 3.2 Finite element spaces and degrees of freedom

We use continuous and discontinuous finite elements for the discretization of the momentum equation as well as the constitutive equations and advection problems, respectively. On the reference element $\hat{T}$, we define two sets of basis functions. The dG-
basis functions that we employ are given by

$$\begin{aligned}
&\Psi_1(\hat{\mathbf{x}}) := 1 && \Psi_2(\hat{\mathbf{x}}) := \hat{\mathbf{x}}_1 - 1/2 \\
&\Psi_3(\hat{\mathbf{x}}) := \hat{\mathbf{x}}_2 - 1/2 && \Psi_4(\hat{\mathbf{x}}) := (\hat{\mathbf{x}}_1 - 1/2)(\hat{\mathbf{x}}_2 - 1/2) \\
&\Psi_5(\hat{\mathbf{x}}) := (\hat{\mathbf{x}}_1 - 1/2)^2 - 1/12, && \Psi_6(\hat{\mathbf{x}}) := (\hat{\mathbf{x}}_2 - 1/2)^2 - 1/12 \\
&\Psi_7(\hat{\mathbf{x}}) := (\hat{\mathbf{x}}_2 - 1/2)\left((\hat{\mathbf{x}}_1 - 1/2)^2 - 1/12\right), && \Psi_8(\hat{\mathbf{x}}) := (\hat{\mathbf{x}}_1 - 1/2)\left((\hat{\mathbf{x}}_2 - 1/2)^2 - 1/12\right)
\end{aligned} \tag{11}$$

These basis functions are orthogonal, i.e. $\int_{\hat{T}} \Psi_i \Psi_j \, d\mathbf{x} = \delta_{ij}$. Second, we use degree $r$ tensor product Lagrange finite element basis functions

$$\Phi_{(r+1)l+k}^{(r)}(\hat{\mathbf{x}}) := \xi_k^{(1)}(\mathbf{x}_1)\xi_l^{(1)}(\mathbf{x}_2), \quad k,l = 1,\dots,r+1. \tag{12}$$

The one-dimensional basis functions (for $r=1$ and $r=2$) are given by

$$\xi_0^{(1)}(\hat{x}) := 1-\hat{x}, \ \xi_1^{(1)}(\hat{x}) := \hat{x}, \quad \xi_0^{(2)}(\hat{x}) := (1-\hat{x})(1-2\hat{x}), \ \xi_1^{(2)}(\hat{x}) := 4\hat{x}(1-\hat{x}), \ \xi_2^{(2)}(\hat{x}) := \hat{x}(2\hat{x}-1). \tag{13}$$



All basis functions are mapped from the reference element $\hat{T}$ onto the mesh elements of $\mathcal{T}_h$.

We define continuous finite element spaces $V_h^{(r)}$, where $r = 1, 2$ is the degree, and spaces $W_h^{(s)}$ associated with the discontinuous finite elements, where $s = 1, 2, \ldots$ is the number of local basis functions,

$$
\begin{aligned}
V_h^{(r)} &= \left\{ \phi \in C(\bar{\Omega}) : \phi\big|_T \in \operatorname{span}\left\{ \Phi_k^{(r)} \circ \mathbf{T}_T^{-1},\ k = 1, \ldots, (r+1)^2 \right\} \quad, \forall T \in \mathcal{T}_h \right\} \\
W_h^{(s)} &= \left\{ \psi \in L^2(\Omega) : \psi\big|_T \in \operatorname{span}\left\{ \Psi_k \circ \mathbf{T}_T^{-1},\ 1, \ldots, s \right\} \quad, \forall T \in \mathcal{T}_h \right\}.
\end{aligned}
\tag{14}
$$

Locally on each mesh element $T \in \mathcal{T}_h$, the tracer $H_h \in W_h^{(s)}$ and velocity $\mathbf{v}_h \in V_h^{(r)}$ are therefore desribed by the linear combinations of the basis functions

$$
\hat{H}_T(\hat{\mathbf{x}}) := H_h\big(\mathbf{T}_T(\hat{\mathbf{x}})\big) = \sum_{j=1}^{s} H_{T,j} \Psi_j(\hat{\mathbf{x}}), \quad \hat{\mathbf{v}}_T(\hat{\mathbf{x}}) := \mathbf{v}_h\big(\mathbf{T}_T(\hat{\mathbf{x}})\big) = \sum_{j=1}^{N_{loc}^{cG}} \mathbf{v}_{T,j} \Phi_k(\hat{\mathbf{x}}),
\tag{15}
$$

where $N_{loc}^{cG} = (r+1)^2$ is the local number of unknowns in each element. An analogous representation holds for the second tracer. Finally, by $(\cdot, \cdot)_T$ and $\langle \cdot, \cdot \rangle_e$ we denote $L^2$-scalar products

$$
(\phi, \psi)_T = \int_T \phi(\mathbf{x}) \psi(\mathbf{x}) \, d\mathbf{x}, \quad \langle \phi, \psi \rangle_e = \int_e \phi(\mathbf{x}) \psi(\mathbf{x}) \, ds,
$$

on the elements $T_{i,j}$ and the edges $e_{i,j}^{\{x,y\}}$, respectively.

### 3.3 Discontinuous Galerkin advection scheme

We begin by describing the discretization of the advection equation

$$
\partial_t H + \operatorname{div}\big(\mathbf{v} H\big) = 0.
$$

for the tracer $H : \Omega \to \mathbb{R}$. We follow the notation of (Di Pietro and Ern, 2012, Chapter 3).

The temporal discretization will be by explicit Runge-Kutta schemes and spatially $H_h \in W_h^{(s)}$, see Eq. 14 and Eq. 15. The discretiation is based on the standard upwind formulation

$$
\sum_{T \in \mathcal{T}_h} \partial_t (H_h, \psi)_T - (H_h \mathbf{v}, \nabla \psi)_T + \sum_{e \in \mathcal{T}_h} \langle \{\!\{ H_h \}\!\}, \mathbf{v} \cdot \boldsymbol{n}_e [\![ \psi ]\!] \rangle_e + \frac{1}{2} \langle |\mathbf{v} \cdot \boldsymbol{n}_e| \cdot [\![ H_h ]\!], [\![ \psi ]\!] \rangle_e = 0.
\tag{16}
$$

By $\{\!\{ H_h \}\!\}\big|_e$ we denote the average of the dG function $H_h$ on an edge $e = \partial T_1 \cap \partial T_2$ between the two elements $T_1, T_2$ and by $[\![ H_h ]\!]\big|_e$ the jump over this edge, i.e.

$$
\{\!\{ H_h \}\!\}\big|_e = \frac{1}{2}\big( H_h\big|_{T_1} + H_h\big|_{T_2} \big), \quad [\![ H_h ]\!]\big|_e = H_h\big|_{T_1} - H_h\big|_{T_2}.
$$

The upwind scheme can be written in matrix-vector notation as

$$
\mathbf{M} \partial_t H_h = \mathbf{A}(\mathbf{v}_h) H_h,
$$



where $\mathbf{M}$ is the dG-mass matrix in $W_h^{(s)}$, which is block-diagonal with blocks of size $s \times s$, and where $\mathbf{A}(\mathbf{v}_h)$ gathers all remaining terms of Eq. (16) which are all linear in $H_h$. The equation is discretized in time by standard explicit Runge-Kutta methods on the advection time mesh in Eq. (7).

   For dG(0) with space $W_h^{(1)}$, the discretization is equivalent to the usual finite volume upwind scheme since the per element term $(H_h\mathbf{v}, \nabla\psi)$ vanishes for all $\psi \in W_h^{(1)}$ as $\psi|_T$ is constant on $T$. The advantage of using higher order methods will become

clear in Sec. 5.3.3.

### 3.4   Discretizing the momentum equation

The coupled advection and momentum equation system in Eq. 1 is decoupled in a partitioned iteration by performing the advection step and then solving the momentum equation. The momentum equation is approximated with an mEVP solver, which can be considered as a pseudo time-stepping scheme for the implicit backward Euler discretization of the VP formulation,

(e.g., see Lemieux et al. (2012); Bouillon et al. (2013)). We introduce the iterates $\mathbf{v}_n^{(p)}$ and $\boldsymbol{\sigma}_n^{(p)}$ for $p = 0, 1, \ldots, N_{\mathrm{mEVP}}$ with $\mathbf{v}_n^{(0)} := \mathbf{v}_{n-1}$ and $\boldsymbol{\sigma}_n^{(0)} := \boldsymbol{\sigma}_{n-1}$ in which case the update can be written as

$$(1+\alpha)\boldsymbol{\sigma}_n^{(p)} = \alpha\boldsymbol{\sigma}_n^{(p-1)} + \boldsymbol{\sigma}(\mathbf{v}_n^{(p-1)}),$$

$$\Big((1+\beta)\rho_{ice}H_n + \Delta t A_n C_o \rho_o \|\mathbf{v}_o - \mathbf{v}_n^{(p-1)}\|_2\Big)\mathbf{v}_n^{(p)} = \rho_{ice}H_n\big(\mathbf{v}_{n-1} + \beta\mathbf{v}_n^{(p-1)}\big) + \Delta t\big(\operatorname{div}\boldsymbol{\sigma}_n^{(p)} + \tilde{F}(\mathbf{v}_n^{(p-1)})\big). \tag{17}$$

The forcing term $F(\mathbf{v}_n^{(p)})$ in Eq. (2) is split into explicit and implicit parts. The ocean forcing term is considered implicitly, which helps the stability of the scheme, and the remaining explicit terms on the right hand side are

$$\tilde{F}(\mathbf{v}_n^{(p-1)}) := A_n\Big(C_o\rho_o\|\mathbf{v}_o - \mathbf{v}_n^{(p-1)}\|_2 \cdot \mathbf{v}_o + C_a\rho_a\|\mathbf{v}_a\|_2 \cdot \mathbf{v}_a\Big) + \rho_{ice}H_n f_c \boldsymbol{e}_z \times (\mathbf{v}_o - \mathbf{v}_n^{(p-1)}). \tag{18}$$

   The parameters $\alpha$ and $\beta$ in Eq. 17 control the stability but also the speed of convergence of the mEVP-iteration to the VP-solution whereas the number of steps $N_{\mathrm{mEVP}}$ controls the accuracy. We refer the reader to Kimmritz et al. (2016) for a discussion on this.

   A mixed finite element approach is used for the spatial discretization of Eqs. (17)-(18) with continuous finite elements for

the momentum equation and discontinuous ones for the stress update. This yields

$$(1+\alpha)\big(\boldsymbol{\sigma}_n^{(p)}, \Psi_h\big) = \alpha\big(\boldsymbol{\sigma}_n^{(p-1)}, \Psi_h\big) + \big(\boldsymbol{\sigma}(\mathbf{v}_n^{(p-1)}), \Psi_h\big),$$

$$\Big(\big((1+\beta)\rho_{ice}H_n + \Delta t A_n C_o \rho_o \|\mathbf{v}_o - \mathbf{v}_n^{(p-1)}\|_2\big)\mathbf{v}_n^{(p)}, \Phi_h\Big) = \Big(\rho_{ice}H_n\big(\mathbf{v}_{n-1} + \beta\mathbf{v}_n^{(p-1)}\big) + \Delta t\tilde{F}(\mathbf{v}_n^{(p-1)}), \Phi_h\Big)$$

$$- \Delta t\Big(\boldsymbol{\sigma}_n^{(p)}, \nabla\Phi_h\Big) \tag{19}$$

for test functions

$$\Psi \in \mathbf{W}_h := [W_h^{(s)}]^{2\times 2, sym} := \{\boldsymbol{\sigma} \in L^2(\Omega)^{2\times 2}, \boldsymbol{\sigma} = \boldsymbol{\sigma}^T, \sigma_{ij} \in W_h^{(s)}, i, j = 1, 2\}, \quad \Phi_h \in \mathbf{V}_h := [V_h^{(r)}]^2. \tag{20}$$

Compatibility of the velocity and stress spaces is important for the stability of the coupled iteration, see Sect. 5.2.3 for an

example of possible instabilities. Stress spaces that are too small do not provide sufficient control of the term $(\boldsymbol{\sigma}(\mathbf{v}), \Psi)$




in Eq. (19) and lead to oscillatory stresses, see the upper right plot in Fig. 13. The problem is related to the control of the energy and in a mixed formulation the spaces $\mathbf{V}_h$ and $\mathbf{W}_h$ must in particular satisfy the Babuška-Brezzi condition (Ern and Guermond, 2021, Theorem 49.13) for well-posedness. For a simplified linear equation, this condition would mean that there exists a constant $\gamma > 0$ such that

$$\inf_{\mathbf{\Phi} \in \mathbf{V}_h} \sup_{\mathbf{\Psi} \in \mathbf{W}_h} \frac{(\mathbf{\Psi}, \nabla\mathbf{\Phi})_\Omega}{\|\nabla\mathbf{\Phi}\| \cdot \|\mathbf{\Psi}\|} \geq \gamma > 0. \tag{21}$$

This condition can easily be satisfied if for every $\mathbf{v}_h \in \mathbf{V}_h$ from the cG-velocity space it holds that

$$\frac{1}{2}\left(\nabla\mathbf{v}_h + \nabla\mathbf{v}_h^T\right) \in \mathbf{W}_h := [W_h^{(s)}]^{2 \times 2, sym}. \tag{22}$$

Then, for any $\mathbf{\Phi} = \mathbf{v}_h$ we choose $\mathbf{\Psi}$ as Eq. (22) and get, using the symmetry of the inner product

$$\frac{\left(\nabla\mathbf{v}_h + \nabla\mathbf{v}_h^T, \nabla\mathbf{v}_h\right)_\Omega}{\|\nabla\mathbf{v}_h + \nabla\mathbf{v}_h^T\|_\Omega \|\nabla\mathbf{v}_h\|_\Omega} = \frac{\left(\nabla\mathbf{v}_h + \nabla\mathbf{v}_h^T, \frac{1}{2}\left(\nabla\mathbf{v}_h + \nabla\mathbf{v}_h^T\right)\right)_\Omega}{\|\nabla\mathbf{v}_h + \nabla\mathbf{v}_h^T\|_\Omega \|\nabla\mathbf{v}_h\|_\Omega} = \frac{1}{2}\frac{\|\nabla\mathbf{v}_h + \nabla\mathbf{v}_h^T\|_\Omega}{\|\nabla\mathbf{v}_h\|_\Omega} \geq \frac{c_K}{2},$$

where $c_K > 0$ is the constant of Korn's inequality (Ern and Guermond, 2021, Theorem 42.9 and 42.10). We therefore require that the spaces $\mathbf{W}_h$ and $\mathbf{V}_h$ always allow for choosing the stress test-function $\mathbf{\Psi} \in \mathbf{W}_h$ as the symmetric velocity gradient, Eq. (22). To be precise, the degree $s$ has to be chosen such that the symmetric gradient of the discrete velocity is part of the stress space. On quadrilateral elements, the continuous finite element basis is not the pure polynomial basis $P^{(r)}$ but it includes the additional mixed terms $xy$ for $r = 1$ and $x^2y, xy^2, x^2y^2$ for $r = 2$. Hence, the gradient space must also be enriched. For linear elements with $r = 1$ the condition in Eq. (22) requires $s = 3$ and for quadratic velocities with $r = 2$ we must take $s = 8$ in Eqs. (19)-(20). This update involves the inversion of the mass matrix of $W_h^{(s)}$. The matrix is block-diagonal with block-size $s \times s$ so that in the cG(2)-case with $s = 8$ the costs for the inversion are substantial. Sec. 4.2 describes our approach for an efficient implementation.

The momentum equation is discretized with continuous finite elements in the discrete space $V_h$. All zero-order terms in the momentum equation, Eq. (19), are evaluated node-wise and no integration is required. Adding the stress, however, requires integration and inversion of the mass matrix of $V_h$. To avoid the inversion, we use mass lumping. The evaluation of the momentum equation's right-hand side in Eq. (19) then becomes

$$\mathbf{v}_{n,i}^{(p)} = \left(1 + \beta\right)\rho_{ice} H_{n,i} + \Delta t A_{n,i} C_o \rho_o \|\mathbf{v}_{o,i} - \mathbf{v}_{n,i}^{(p-1)}\|_2\right)^{-1} \cdot$$
$$\cdot \left(\rho_{ice} H_{n,i}\left(\mathbf{v}_{n-1,i} + \beta\mathbf{v}_{n,i}^{(p-1)}\right) + \Delta t \tilde{F}(\mathbf{v}_{n,i}^{(p-1)}) - \mathbf{M}_{l,ii}^{-1}\Delta t\left(\boldsymbol{\sigma}_n^{(p)}, \nabla\Phi_i\right)_\Omega\right), \quad i = 1, \ldots, N_{cG} \tag{23}$$

where $\mathbf{M}_l$ is the lumped mass matrix in the cG-velocity space. The implicit terms are handled analogously. The integration of the stresses against the gradient of the test function is a non-local operation coupling adjacent degrees of freedom. All other operations, like computing $\tilde{F}(\mathbf{v}_n^{(p-1)})$, are fully decoupled and can be processed node-wise in parallel.

**Remark 1** (Optimality of the velocity-stress discretization). *On triangular meshes, $V_h^{(r)}$ would be the spaces of piecewise polynomials of degree $r$. In this case, the optimal dG-stress space would be the space of piecewise constants, i.e. $W_h^{(1)}$ in our*



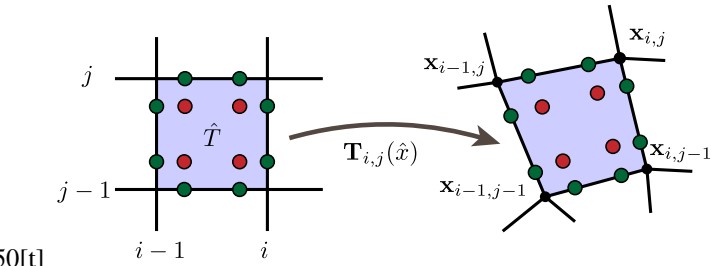

**Figure 2.** The numerical quadrature nodes $\hat{\chi}_q$ are defined on the reference element $\hat{T} = (0, 1)^2$ and mapped to the real mesh elements $T \in \mathcal{T}_h$ via $\chi_q := \mathbf{T}(\hat{\chi}_q)$. We show a 2-point Gauss rule (of degree 4) with 2 points on each edge and 4 points in the element.

50[t]

*notation in the case $r = 1$ and the space of piecewise linears $W_h^{(3)}$ for $r = 2$. The choice given in Eq. (19)-(20) appears highly inefficient as the local basis has 3 instead of 1 unknowns for $r = 1$ and 8 instead of 3 for $r = 2$. However, a triangular mesh with the same number of velocity unknowns has twice the number of elements as a quadrilateral mesh. Hence, $r = 1$ has 2 unknowns on triangles and $r = 2$ brings 6 unknowns compared to $s = 3$ and $s = 8$ in the case of quadrilateral meshes. This*
*means that the difference in effort is less dramatic than it appears on first sight.*

### 3.5 Numerical quadrature

In the parametric finite element setup, all integrals appearing in the advection scheme in Eq. (16) and the weak formulation of the mEVP iteration in Eq. (17) must be evaluated on the reference element $\hat{T}$ and, in case of the upwind scheme, also on the reference edge $\hat{e} = (0, 1)$ since the basis functions are defined on $\hat{T}$. For the different terms of Eq. (16) it holds

$$\partial_t (H_h, \psi)_T = (J_T \partial_t \hat{H}_h, \Psi)_{\hat{T}} \qquad (H_h \mathbf{v}, \nabla \psi)_T = (\hat{H}_h \hat{\mathbf{v}}, J_T \hat{\nabla} \mathbf{T}_T^{-T} \hat{\nabla} \Psi)_{\hat{T}}$$
$$\langle \{\{H_h\}\}, (\mathbf{v} \cdot \boldsymbol{n}_e) [\![\psi]\!] \rangle_e = \langle J_e \{\{\hat{H}_h\}\}, \widehat{(\mathbf{v} \cdot \boldsymbol{n}_e)} [\![\Psi]\!] \rangle_{\hat{e}} \qquad \frac{1}{2} \langle |\mathbf{v} \cdot \boldsymbol{n}_e| \cdot [\![H_h]\!], [\![\psi]\!] \rangle_e = \frac{1}{2} \langle J_e | \widehat{\mathbf{v} \cdot \boldsymbol{n}_e} | \cdot [\![\hat{H}_h]\!], [\![\Psi]\!] \rangle_{\hat{e}}$$
(24)

where $\hat{H}_h$ and $\hat{\mathbf{v}}$ are the functions on the reference element that by Eq. (15) are associated with $H_h$ and $\mathbf{v}$ on the element $T$, and analogously for the edge terms. The reference element map $\mathbf{T}_T$ dependent terms in Eq. 24 are the Jacobian $\hat{\nabla} \mathbf{T}_T : \hat{T} \to \mathbb{R}^{2 \times 2}$ and its determinant $J_T = \det(\hat{\nabla} \hat{\mathbf{T}}_T)$. Since $\mathbf{T}_T$ is bi-linear (and not linear), the Jacobian and its determinant are not element-wise constant. However, on the reference edges $\hat{e}$, $\mathbf{T}_T$ is linear such that $e \in \mathcal{T}_h$ are straight and hence $J_e = |e|$ as the reference
element has edge length 1.

The integrals in Eq. (24) are approximated by Gaussian quadrature. For dG(r) ($r = 0, 1, 2$) we use $r + 1$ quadrature points on the edge and $(r + 1)^2$ quadrature points within the elements, see Fig. 2 for an example with two points on the edges and $2 \times 2$ points within the element.

Implementation details are described in Section 3.1. Evaluation of the terms in (24) is numerically costly, mostly due to the
evaluation of the map $\mathbf{T}_T$, the Jacobian $\hat{\nabla} \mathbf{T}_T$, its inverse and the determinant of the Jacobian.





## 4 Efficient parallelizable implementation

In the following paragraphs, we will describe the C++ implementation of the higher-order discretization. A hybrid parallelization approach consisting of distributed memory MPI splitting and local shared memory OpenMP realization is considered. The data is structured such that the implementation also allows to run modules on a GPU.

MPI parallelization builds on a domain decomposition that splits the complete mesh into a balanced number of rectangular subdomains such that the average number of ice-covered elements for each domain is comparable. Each parallel task then operates on a subdomain that is topologically structured into $N_{el} := N_x \times N_y$ elements such as described in Sect. 3.1.

### 4.1 Implementation of continuous and discontinuous finite elements

We start by describing the handling of the data, i.e. the cG- and dG-vectors for each MPI task that is responsible for one
topologically rectangular mesh $\mathcal{T}_h$ consisting of $N_x \times N_y$ elements. A dG-vector $A_h \in W_h^{(s)}$ has $s$ unknowns on each of the $N_{el} = N_x \cdot N_y$ elements and we store such a vector as a $\mathbf{A} \in \mathbb{R}^{N_{el} \times s}$ matrix. The implementation is based on *Eigen* (Guennebaud et al., 2010), a C++ library for linear algebra that heavily relies on C++ templates. In the code, the vector is represented as

```
1:    Matrix<FloatType, Dynamic, s, RowMajor> DGVector<s> A;
```

The first dimension of `A` (number of elements) is dynamic and determined at run-time, which allows us to flexibly handle different subdomain sizes. The second dimension, i.e. the number of components, has degree $s$ and is determined at compile time. This allows for vectorized SIMD processing of computations, see Sect. 5.3.2 for a numerical demonstration.

To provide one example of a frequently used operation, we explain the restriction of a dG(1) function $A_h \in W_h^{(3)}$ (with three local unknowns on $T$) from an element $T$ to one of its edges $e \in \partial T$. Let $T$ have the element-id $i \in \{1, \ldots, N_{el}\}$ and let $e = e_i^x$ be the lower edge in the notation of (9). Then the restriction to the lower edge is realized as

```
1: Vector<FloatType, 2> lower_edge(const DGVector<3>& A, size_t i)
2: { return Vector<FloatType, 2> a_e({A(i,0)-0.5 * A(i,2), A(i,1)}); }
```

Since the restriction does not depend on the specific element $T \in \mathcal{T}_h$, the relations are implemented for the four edges and the different choices of dG-spaces, i.e. for the number of local basis functions, using template specializations. With this, both *Eigen* and the compiler can optimize the computations.

The parametric setup also allows for an efficient restriction of a dG or cG function to the Gauss points. Let $A_h \in W_h^{(6)}$ and let $T \in \mathcal{T}_h$ be again any mesh element with element-id $i \in \{1, \ldots, N_{el}\}$. Assume that we want to evaluate $A_h$ in the $3 \times 3$ Gauss points $\hat{\boldsymbol{\chi}}_q \in (0,1)^2$, cf. Sect. 3.5. It holds $\boldsymbol{\chi}_q^i := \mathbf{T}_T(\hat{\boldsymbol{\chi}}_q)$ and hence

$$A_h(\boldsymbol{\chi}_q^i) = \sum_{l=1}^{6} \mathbf{A}_{i,l} \Psi_l\big(\mathbf{T}_T^{-1}(\boldsymbol{\chi}_q^i)\big) = \sum_{l=1}^{6} \mathbf{A}_{i,l} \hat{\Psi}_l(\hat{\boldsymbol{\chi}}_q) =: \mathbf{A}_i^i. \tag{25}$$

That is, by working with the pulled back function $\hat{\Psi}_l$ on the reference element, $\hat{\Psi}_l$ only needs to be evaluated on the fixed
points $\hat{\boldsymbol{\chi}}_q$. Furthermore, by the linearity of the basis representation, the mapping of the local coefficients $\mathbf{A}_{i,1}, \ldots, \mathbf{A}_{i,6}$ of the



dG vector to the values of $\mathbf{A}_i^G \in \mathbb{R}^9$ in the 9 Gauss points on the element $T$ can be written as a matrix-vector product

$$\mathbf{A}_i^G = \mathbf{A}_{i,\cdot} \cdot \mathbf{G}_\Psi^{9,6}, \quad [\mathbf{G}_\Psi^{9,6}]_{l,q} = \hat{\Psi}_l(\hat{\boldsymbol{\chi}}_q). \tag{26}$$

with a fixed matrix $\mathbf{G}_\Psi^{9,6} \in \mathbb{R}^{9\times 6}$. The matrices $\mathbf{G}_\Psi^{q,s}$ for possible dG-degrees with $s$ local unknowns and for supported choices of the Gauss quadrature rule with $q$ points are pre-computed and directly inlined into the code to allow for an optimization by

*Eigen* and the compiler. The matrices $\mathbf{G}_\Psi^{s,q}$ and similar code are auto-generated by Python scripts to allow for easy extension.

Another challenge for an efficient implementation is the evaluation of the integrals that are required to determine the viscosity within the VP-model (4) in the mEVP iteration, see Eq. (19),

$$\zeta = \frac{P^\star \cdot H_h \cdot \exp\left(-C(1-A_h)\right)}{\sqrt{\Delta_{\min}^2 + \text{tr}(\boldsymbol{\epsilon}_h)^2 + \frac{2}{e^2}\boldsymbol{\epsilon}_h' : \boldsymbol{\epsilon}_h'}} = \frac{P^\star \cdot H_h \cdot \exp\left(-C(1-A_h)\right)}{\sqrt{\Delta_{\min}^2 + \frac{5}{4}(\boldsymbol{\epsilon}_{11} + \boldsymbol{\epsilon}_{22})^2 + \frac{3}{2}\boldsymbol{\epsilon}_{11}\boldsymbol{\epsilon}_{22} + \boldsymbol{\epsilon}_{12}^2}}.$$

With $i \in \{1, \ldots, N_T\}$ again denoting the element index, the following example illustrates the evaluation of the viscosities in

the 9 Gauss points in the case of biquadratic velocities, a strain tensor with 8 local unknowns, i.e. $E_h \in [W_h^{(8)}]^{2\times 2, sym}$, and tracers discretized as dG(1)-functions in $W_h^{(3)}$.

```
1: const Array<9> Ag   = A.row(i)   * Gpsi<9,3>;  // restrict k-th element to Gauss points
2: const Array<9> Hg   = H.row(i)   * Gpsi<9,3>;  // restrict ice height to Gauss points
3: const Array<9> E11g = E11.row(i) * Gpsi<9,8>;  // restrict strain tensor to Gauss points
4: const Array<9> E12g = E12.row(i) * Gpsi<9,8>;  // restrict strain tensor to Gauss points
5: const Array<9> E22g = E22.row(i) * Gpsi<9,8>;  // restrict strain tensor to Gauss points
6:
7: const Array<9> zeta = Pstar * Hg * (-C * (1-Ag)).exp() /
8:    (Dmin*Dmin + 1.25 * (E11g+E22g).square() + 1.5 * E11g * E22g + E12g.square()).sqrt();
```

The above implementation is close to the mathematical notation which simplifies the implementation of model variations. Long expressions such as those in the last line also allow *Eigen* to vectorize operations efficiently.

## 4.2 Evaluation of the weak formulations on parametric meshes

A substantial part of the computational effort is due to the mapping of the reference element $\hat{T}$ onto the mesh elements $T \in \mathcal{T}_h$, compare Sect. 3.1 and Sect. 3.5. We discuss the details of an efficient implementation for one specific term in the mEVP momentum equation (19), namely the evaluation of $(\boldsymbol{\sigma}_n^{(p)}, \nabla\boldsymbol{\Phi}_i)_\Omega$ whose discretization has already been given in Eq. (23). In the following, we will omit all indices referring to the time step and the mEVP iteration count.

At the heart of $(\boldsymbol{\sigma}_n^{(p)}, \nabla\boldsymbol{\Phi}_i)_\Omega$ is the integration of the symmetric stress tensor multiplied with the gradient of the (vector-

valued) test function $\boldsymbol{\Phi}_i = (\Phi_i^x, \Phi_i^y)$. Pulling this term back from a triangle $T \in \mathcal{T}_h$ onto the reference element $\hat{T}$, we obtain

$$(\boldsymbol{\sigma}, \nabla\boldsymbol{\Phi}_i)_T = \int_{\hat{T}} \det\left(\hat{\nabla}\mathbf{T}_T(\hat{x})\right)\hat{\nabla}\boldsymbol{\Phi}_i(\hat{x})\hat{\nabla}\mathbf{T}_T^{-T}(\hat{x}) : \hat{\boldsymbol{\sigma}}(\hat{x})\,\mathrm{d}\hat{x}, \quad i = 1, \ldots, N_{loc}^{\text{cG}}. \tag{27}$$





Here, $N_{loc}^{\mathrm{cG}} := (r+1)^2$ is the local number of cG-degrees of freedom and $\mathbf{A} : \mathbf{B} := \sum_{i,j} \mathbf{A}_{ij} \mathbf{B}_{ij}$ is the full contraction of rank-2 tensors. Locally on the element $T \in \mathcal{T}_h$, symmetric stresses $\boldsymbol{\sigma} \in [W_h^{(s)}]^{2 \times 2, sym}$ and the element map's gradient $\hat{\nabla} \mathbf{T}_T$ are given

in the dG- and cG-basis as

$$
\boldsymbol{\sigma}(\hat{x})\Big|_T = \sum_{j=1}^{s} \underbrace{\begin{pmatrix} \sigma_{T,j}^{11} & \sigma_{T,j}^{12} \\ \sigma_{T,j}^{12} & \sigma_{T,j}^{22} \end{pmatrix}}_{= \boldsymbol{\sigma}_{T,j}} \Psi_j(\hat{x}), \quad \hat{\nabla} \mathbf{T}_T(\hat{\mathbf{x}}) = \sum_{k=1}^{4} \underbrace{\begin{pmatrix} x_{T,k}^1 \\ x_{T,k}^2 \end{pmatrix}}_{= \mathbf{x}_{T,k}} \underbrace{\begin{pmatrix} \partial_{\hat{x}} \Phi_k(\hat{\mathbf{x}}) & \partial_{\hat{y}} \Phi_k(\hat{\mathbf{x}}) \end{pmatrix}}_{= \hat{\nabla} \Phi_k(\hat{\mathbf{x}})^T} \tag{28}
$$

with the $\mathbf{x}_{T,k} = (x_{T,k}^1, x_{T,k}^2) \in \mathbb{R}^2$ being the four corner nodes of the element $T$. Approximating Eq. (27) and Eq. (28) by Gauss quadrature with $n_Q$ points $\hat{\mathbf{x}}_1, \ldots, \hat{\mathbf{x}}_{n_Q} \in \hat{T}$ and weights $\omega_1, \ldots, \omega_{n_Q}$ yields

$$
(\boldsymbol{\sigma}, \nabla \boldsymbol{\phi}_i)_T \approx \sum_{j=1}^{s} \underbrace{\sum_{q=1}^{n_Q} \omega_q \det\left(\hat{\nabla} \mathbf{T}(\hat{\mathbf{x}}_q)\right) \Psi_j(\hat{\mathbf{x}}_q) \hat{\nabla} \Phi_i(\hat{\mathbf{x}}_q) \hat{\nabla} \mathbf{T}(\hat{\mathbf{x}}_q)^{-T} : \boldsymbol{\sigma}_j}_{=: \mathbf{X}_{i,j}}, \quad i = 1, \ldots, N_{loc}^{\mathrm{cG}}. \tag{29}
$$

The computational effort of the above equation is substantial. The Jacobian $\hat{\nabla} \mathbf{T}_T$ needs to be assembled $n_Q \cdot s \cdot N_{loc}^{\mathrm{cG}}$-times, cf. Eq. (29), and its inverse and determinant need to be computed. For the second order case cG(2) with $n_Q = 9$, $N_{loc}^{\mathrm{cG}} = 9$ and $s = 8$, more than $15,000$ floating point operations are required on each element.

The entries of the $2 \times 2$-matrices $\mathbf{X}_{i,j}$, however, do not depend on the solution but only on the mesh elements $T \in \mathcal{T}_h$. A closer analysis further reveals that $\mathbf{X}_{i,j}^1 := \mathbf{X}_{i,j}^{11} = \mathbf{X}_{i,j}^{21}$ and $\mathbf{X}_{i,j}^2 := \mathbf{X}_{i,j}^{12} = \mathbf{X}_{i,j}^{22}$. Hereby, the complete scalar product with

Gauss approximation is evaluated as

$$
\left( \boldsymbol{\sigma}, \nabla \begin{pmatrix} \Phi_i^x \\ 0 \end{pmatrix} \right)_T \approx \mathbf{X}_T^1 \sigma_T^{11} + \mathbf{X}_T^2 \sigma_T^{12}, \quad \left( \boldsymbol{\sigma}, \nabla \begin{pmatrix} 0 \\ \Phi_i^y \end{pmatrix} \right)_T \approx \mathbf{X}_T^1 \sigma_T^{12} + \mathbf{X}_T^2 \sigma_T^{22}, \tag{30}
$$

with matrices $\mathbf{X}_T^1, \mathbf{X}_T^2 \in \mathbb{R}^{N_{loc}^{\mathrm{cG}} \times s}$. The computational effort shrinks then to $4 N_{loc}^{\mathrm{cG}} \times s^2$ operations, which in the case of cG(2) aounts to about $2\,300$ operations. The matrices $\mathbf{X}_T^1$ and $\mathbf{X}_T^2$ can be precomputed and stored for each mesh element. Their small size makes them, furthermore, amenable for efficient caching although additional storage is needed. Section 5.3.3 presents a

numerical study on the effective performance of the alternatives, i.e. using precomputed matrices or computation of all terms on the fly.

The same technique can be applied to all further terms of (17). For some of them the computational savings of precomputing per element terms are even more substantial. This is in particular true if the inverse of the block-diagonal dG-mass matrix is required, such as in the mEVP iteration (19).

## 4.3 OpenMP parallelization

In each MPI task, only topologically regular rectangular meshes are considered that consist of $N_{el} := N_x \times N_y$ elements. As the complete numerical workflow is based on explicit integrators, OpenMP parallelization is easily realized. Depending on the specific task, a different coloring of the mesh elements (or mesh edges) is utilized to avoid any memory conflicts:





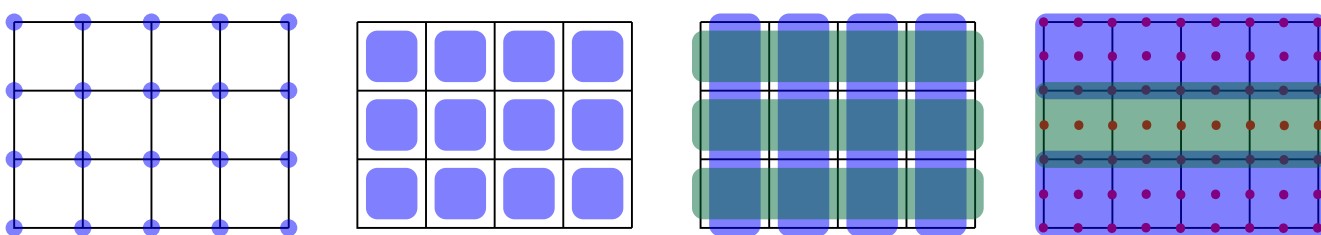

**Figure 3.** Parallel processing of the vectors on a small mesh with 12 elements, $N_x = 4$ and $N_y = 3$. All blocks of one color can be processed in parallel without memory conflicts. From left to right: node-wise operations, local element-wise operations, edge-wise operations on $e^x$-edges (blue) and $e^y$-edges (green), operations writing on biquadratic cG(2)-vectors.

**Node-wise**  Vector operations (such as sums, entry-wise products, etc.) are parallel with respect to the major index referring to the node.

**Element-wise**  Operations such as the stress-update in Eq. (19) within the mEVP iteration in Eq. (17) are parallel with respect to the mesh element. This also includes the element-wise terms $(H_h \mathbf{v}, \nabla \psi)_T$ of the transport scheme in Eq. (16) where no communication is involved and also the projection of the strain rate tensor from the cG- to the dG-space

$$(J_T \mathbf{E}_h, \Psi)_{\hat{T}} = \frac{1}{2} \big( J_T (\hat{\nabla} \hat{\mathbf{v}} [\hat{\nabla} \mathbf{T}_T]^{-1} + [\hat{\nabla} \mathbf{T}_T]^{-T} \hat{\nabla} \hat{\mathbf{v}}^T), \Psi \big)_{\hat{T}}$$

**Edge-wise**  The edge integrals in Eq. (16) are assembled in two sweeps. First, all horizontal edges $e_x \in \mathcal{T}_h$ are computed

$$\sum_{i_x=1}^{N_x} \sum_{i_y=0}^{N_y} \langle \{\{H_h\}\}, \mathbf{v} \cdot \boldsymbol{n}_e [\![\psi]\!] \rangle_{e^x_{i_x, i_y}} + \frac{1}{2} \langle |\mathbf{v} \cdot \boldsymbol{n}_e| \cdot [\![H_h]\!], [\![\psi]\!] \rangle_{e^x_{i_x, i_y}}$$

and the outer (in $x$-direction) is run in parallel as the integral on an edge $e^x_{i_x, i_y}$ will affect the two elements atop and below it. Then, a second sweep, parallelized in $y$-direction, performs the computation for the $e^y$-edges.

When updating cG-vectors, e.g. in the stress update (cf. Eq. (23)), more care is required. We use a row-wise coloring of the elements and perform the update in two sweeps. Fig. 3 summarises the parallel processing of the mesh.

**Remark 2** (Towards GPU acceleration). *Our finite element discretization requires a large number of per element computations with a substantial amount of computations for each one. Furthermore, the computational costs increase substantially with the order, cf. Sec. 4.2. Only local coupling between adjacent elements thereby exists since an explicit time stepping and mEVP iterations are used. This makes the problem well suited for a GPU parallelization where thousands of independent computations are required to fully utilize a state-of-the-art GPU and even more when multiple GPU are combined in a node. The current implementation has already been designed with a GPU implementation in mind. Its realization is planned as a next step.*





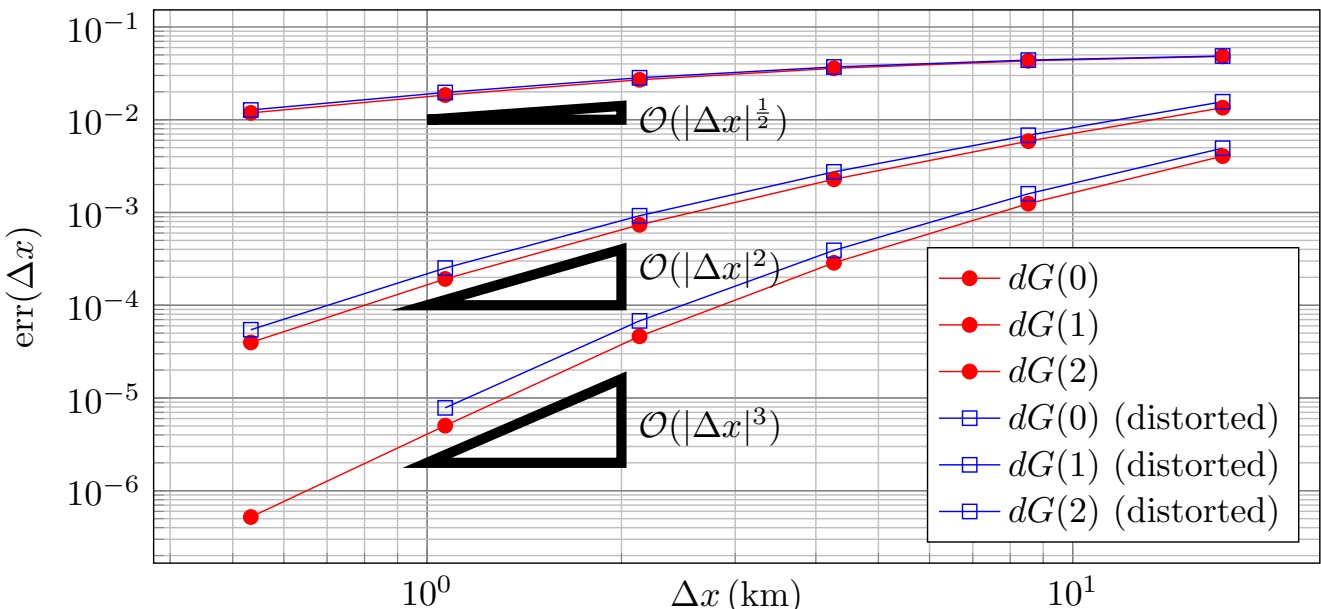

**Figure 4.** Advection testcase I: Convergence rates on uniform (red) and distorted (blue) meshes.

## 5 Numerical experiments

In this section we will present a set of experiments to validate our discretization. We will thereby first only study the accuracy of the advection before considering the full mEVP scheme.

### 5.1 Validating the higher-order transport scheme

### 5.1.1 Advection testcase I: transport of a initially smooth bump

On the domain $\Omega = (0, L_x) \times (0, L_y)$ with $L_x = 409\,600$ and $L_y = 512\,000$ we advect the initially smooth bump

$$H_{\text{in}}(\mathbf{x}) = \begin{cases} \exp\left(-\frac{1}{1-r(\mathbf{x})}\right) & r(\mathbf{x}) < 1 \\ 0 & r(\mathbf{x}) \geq 0 \end{cases}, \quad r(\mathbf{x}) = 40\left\|\frac{\mathbf{x}}{L_x} - \left(\frac{1}{4}, \frac{1}{2}\right)^T\right\|^2$$

with the stationary, rotational velocity field

$$\mathbf{v}(\mathbf{x}) = \frac{\pi}{L_x} \begin{pmatrix} 2\mathbf{x}_2 - L_x \\ L_x - 2\mathbf{x}_1 \end{pmatrix}.$$

The problem is run in the time interval $T = [0, L_x]$ such that one complete revolution of the bump is performed. We compute the test case on a sequence of meshes consisting of $N_x^{(l)} \times N_y^{(l)}$ elements and $N_T^{(l)}$ time steps using

$$N_x^{(l)} = 24 \cdot 2^{l-1}, \quad N_y^{(l)} = 26 \cdot 2^{l-1}, \quad N_T^{(l)} = 200 \cdot 2^{l-1} \cdot (r+1)^{-2}, \quad l = 1, 2, \ldots,$$



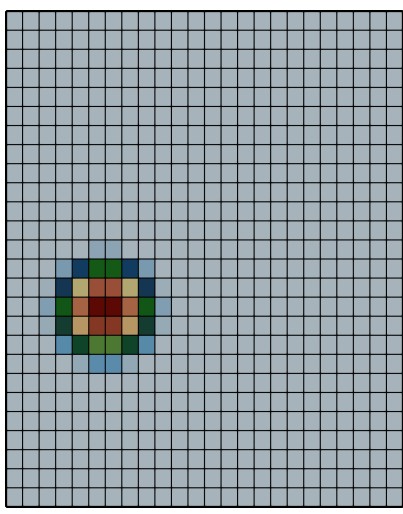 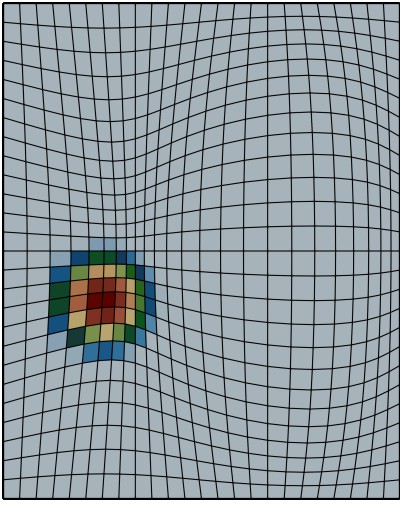

**Figure 5.** Advection testcase I: Visualization of the coarse meshes and the initial $dG(0)$ solution. Left: regular rectangular mesh. Right: distorted parametric mesh.

where $r \in \{0, 1, 2\}$ is the degree of the dG(r) approach. The coarsest discretization consists of $24 \cdot 26 = 624$ elements of approximate size $17\,\text{km} \times 19.6\,\text{km}$ each and a time step $\Delta t = 512\,\text{s}$. This results in a CFL constant lower than $0.5 \cdot (r+1)^{-2}$, which is sufficient for a robust discretization. Next to these uniform rectangular meshes, we use a sequence of distorted meshes to model the effect one encounters in a mesh parametrization of the sphere, see Fig. 5. The nodes $\mathbf{x}_{i,j}$ are in this case given by

$$\mathbf{x}_{i,j} = \begin{pmatrix} \frac{i \cdot L_x}{N_x} + \frac{1}{20} \sin\left(\frac{i \cdot 3\pi}{N_x}\right) \sin\left(\frac{j \cdot \pi}{N_y}\right) \\ \frac{j \cdot L_y}{N_y} + \frac{1}{20} \sin\left(\frac{i \cdot 2\pi}{N_x}\right) \sin\left(\frac{j \cdot 2\pi}{N_y}\right) \end{pmatrix}, \quad \text{for } i = 0, \ldots, N_x, \text{ and } j = 0, \ldots, N_y.$$

Through the periodicity of the domain, the exact solution at time $T = L_x$ equals the initial condition. We measure the scaled $L^2$-error by

$$\text{err} = \frac{1}{L_x} \|H_{h,\Delta t}(T) - H_{\text{in}}\|_{L^2(\Omega)}.$$

The scaling factor $1/L_x$ accounts for the drift-error accumulation that is expected to be dependent on the length of the advection in space.

Fig. 4 shows the convergence behavior for the different meshes and degrees $r$. We observe the expected convergence rate of $\mathcal{O}(|\Delta x|^{\frac{1}{2}})$ for dG(0), cf. (Di Pietro and Ern, 2012, Theorem 3.7) and even super-convergent second order instead of $\mathcal{O}(|\Delta x|^{1+1/2})$ for dG(1) and super-convergent third order instead of $\mathcal{O}(|\Delta x|^{2+1/2})$ for dG(2), cf. (Di Pietro and Ern, 2012, Theorem 3.13). Distortion of the meshes slightly increases the error constant but the convergence order is not affected, as expected.





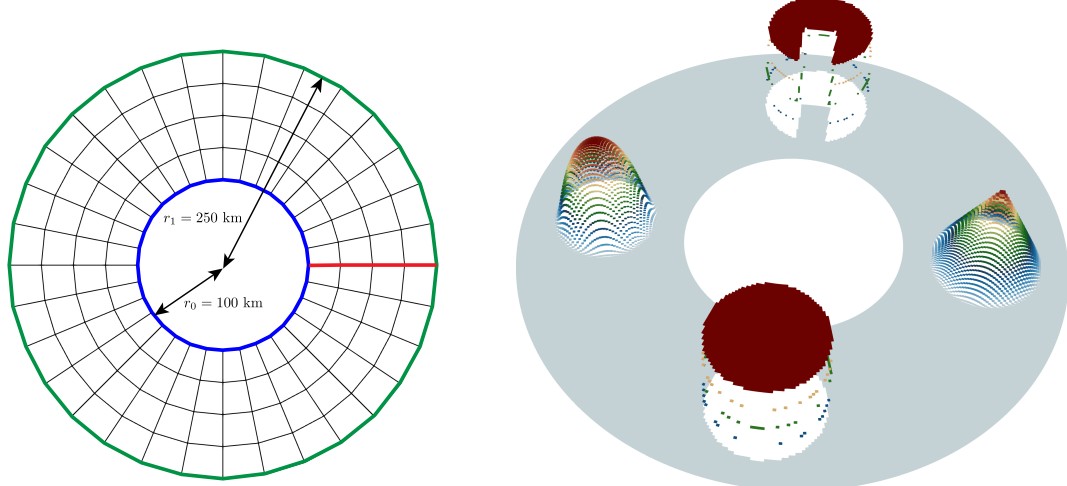

**Figure 6.** Advection testcase II: Computational domain and mesh (left) for the *transport in a ring*. The boundary lines marked in red are the periodic boundaries of the regular reference mesh. On the right we show the initial solution.

### 5.1.2 Advection testcase II: transport in a cicular annulus

The domain of the second test case is a circular annulus with inner radius $r_0 = 100\,\mathrm{km}$ and outer radius $r_1 = 250\,\mathrm{km}$, cf. Fig. 6. The parametric mesh is constructed by mapping a uniform rectangular mesh onto the ring using the map

$$\mathbf{T}(\mathbf{x}_i) := \left(r_0 + (r_1 - r_0)\frac{i_y}{N_y}\right)\begin{pmatrix} \cos\left(\frac{2\pi \cdot i_x}{N_x}\right) \\ -\sin\left(\frac{2\pi \cdot i_x}{N_x}\right) \end{pmatrix}$$

The divergence free stationary velocity field for the transport is given by

$$\mathbf{v}(\mathbf{x}) = \frac{2\pi\,\mathrm{m}}{250\,000\,\mathrm{s}} \cdot \begin{pmatrix} \mathbf{x}_2 \\ -\mathbf{x}_1 \end{pmatrix}.$$

and it moves the initial conditions uniformly along the domain.

    One complete revolution around the annulus is achieved in $T = 250\,000\,\mathrm{s}$. The initial field consists of four objects with different regularity: a smooth $C^\infty$-bump centered at $(-175\,\mathrm{km}, 0\,\mathrm{km})$ of radius $50\,\mathrm{km}$ (on the left), a continuous $C^0$-pyramid centered at $(175\,\mathrm{km}, 0\,\mathrm{km})$ with radius $50\,\mathrm{km}$ (on the right) and two discontinous discs with radius $50\,\mathrm{km}$ at $(0\,\mathrm{km}, -175\,\mathrm{km})$ (on the bottom) and $(0\,\mathrm{km}, 175\,\mathrm{km})$ (on the top). The last one has a "pacman-shaped" omission.

Fig. 7 shows the absolute error between the initial condition and the solution at time $T$. The results verify the superiority of higher-order methods with respect to numerical diffusion. While the finite volume dG(0) discretization is highly diffusive and inaccurate, both dG(1) and dG(2) preserve all shapes well with very small errors for the smooth ones and small, but sharp interface errors for the disc and the pacman-like shape.





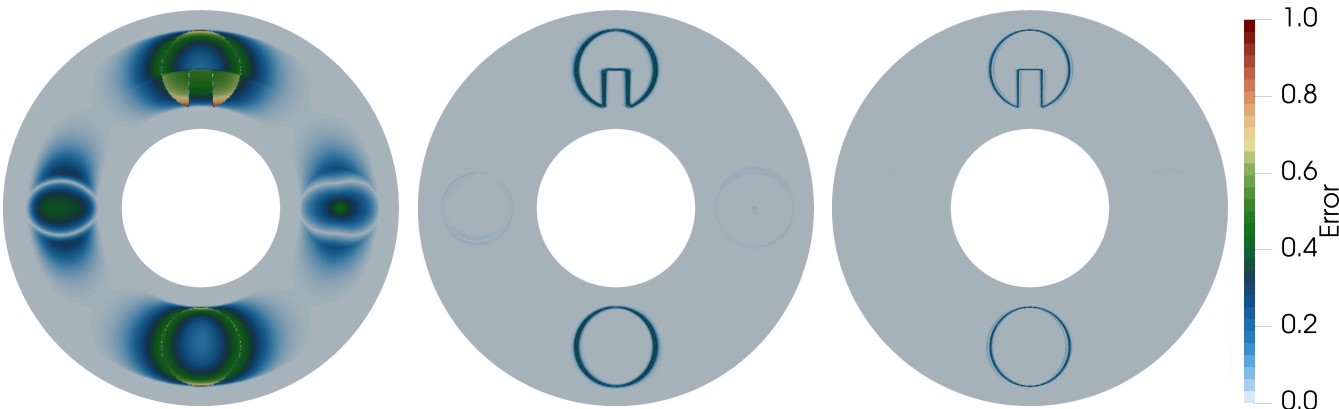

**Figure 7.** Difference (absolute value) between the initial condition and the solution after one revolution for dG(0) (left), dG(1) (middle) and dG(2) (right).

## 5.2 Validating the mEVP framework

To validate the complete dynamical mEVP framework and to assess the higher order discretization, we study the VP benchmark problem that has been introduced in Mehlmann and Richter (2017) and investigated in Mehlmann et al. (2021b) to compare different sea ice realizations with respect to their ability to depict linear kinematic features.

The setup of the benchmark is described in Fig. 8. On the domain of size $512 \, \text{km} \times 512 \, \text{km}$, the ice has an initial concentration of one and an average height of $0.3 \, \text{m}$ with a small amplitude oscillations of $0.005 \, \text{m}$ and with wavelength $105 \, \text{km}$ in $x$- and

$210 \, \text{km}$ in $y$-direction, i.e. we have

$$\mathbf{v}\big|_{t=0} = 0, \quad A(\mathbf{x})\big|_{t=0} = 1, \quad H(\mathbf{x})\big|_{t=0} = 0.3 \, \text{m} + 0.005 \, \text{m} \Big( \sin \big( \frac{6 \mathbf{x}_1}{100 \, \text{km}} \big) + \sin \big( \frac{3 \mathbf{x}_2}{100 \, \text{km}} \big) \Big).$$

The forcing in the benchmark problem consists of a rotational ocean forcing

$$\mathbf{v}_o(\mathbf{x}) = \frac{0.01 \, \text{m} \cdot \text{s}^{-1}}{512 \, \text{km}} \begin{pmatrix} 2 \mathbf{x}_2 - 512 \, \text{km} \\ 512 \, \text{km} - 2 \mathbf{x}_1 \end{pmatrix}$$

and a rotational divergent wind field with a center $\mathbf{m}(t)$ that is moving along the diagonal of the domain

$$\mathbf{v}_a(\mathbf{x}) = \frac{1}{100} \exp \Big( 1 - \frac{|\mathbf{x} - \mathbf{m}(t)|_2}{100 \, \text{km}} \Big) \begin{pmatrix} \cos(\alpha) & \sin(\alpha) \\ -\sin(\alpha) & \cos(\alpha) \end{pmatrix} (\mathbf{x} - \mathbf{m}(t)), \quad \mathbf{m}(t) = \big( 256 \, \text{km} + t \cdot 51.2 \, \text{km} \cdot \text{day}^{-1} \big) \begin{pmatrix} 1 \\ 1 \end{pmatrix}.$$

The different parameters of the VP model and the mEVP iteration are given in Table 1.

Fig. 9 shows the shear $S(\boldsymbol{\epsilon}) := \sqrt{(\epsilon_{11} - \epsilon_{22})^2 + 4 \epsilon_{12}^2}$ at final time $T = 2 \, \text{days}$ on a mesh with $h = 2 \, \text{km}$ spacing. Results are presented for all combinations of the velocity discretization (low order cG(1) and high order cG(2)) and advection schemes dG(0), dG(1) and dG(2). The results are in very good agreement with earlier results for the benchmark (Mehlmann and Richter,



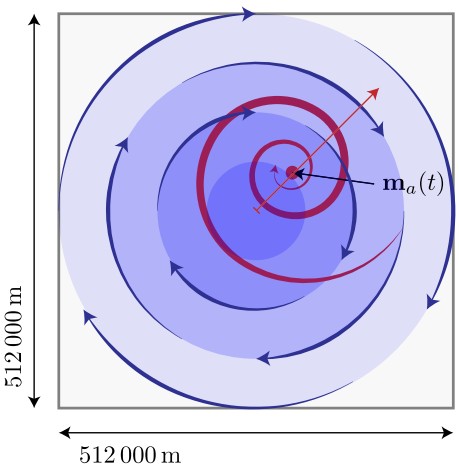

**Figure 8.** Setup of the benchmark problem. Forcing by means of a steady circular ocean current (in blue) and a divergent wind field (in red) that is moving diagonally across the domain (with side length $512\,\mathrm{km}$). Initially the ice has average height $0.3\,\mathrm{m}$ (with a periodic variation) and concentration $A = 1$. The simulation is run for a period of $2\,\mathrm{days}$, where the wind's center travels from the midpoint to the upper right (red arrow).

| Param. | Definition | Value | Param. | Definition | Value |
|---|---|---|---|---|---|
| $\rho_{\mathrm{ice}}$ | Sea ice density | $900\,\mathrm{kg}\cdot\mathrm{m}^{-3}$ | $C_a$ | Air drag | $1.2\cdot 10^{-3}$ |
| $\rho_{\mathrm{a}}$ | Air density | $1.3\,\mathrm{kg}\cdot\mathrm{m}^{-3}$ | $C_o$ | Water drag | $5.5\cdot 10^{-3}$ |
| $\rho_{\mathrm{o}}$ | Water density | $1026\,\mathrm{kg}\cdot\mathrm{m}^{-3}$ | $f_c$ | Coriolis | $1.46\cdot 10^{-4}\,\mathrm{s}^{-1}$ |
| $P^\star$ | Ice strength | $27.5\cdot 10^{3}\,\mathrm{N}\cdot\mathrm{m}^{-2}$ | $C$ | Ice concentration | 20 |
| $e$ | Ellipse ratio | 2 | | | |
| $T$ | time horizon | $2\,\mathrm{days}$ | $\alpha$ | 1st mEVP param. | 1500 |
| $NT_{evp}$ | Subcycling steps | 100 | $\beta$ | 2nd mEVP param. | 1500 |

**Table 1.** Default values of the VP model used to define the benchmark Mehlmann and Richter (2017) as well as default mEVP-parameters used in all numerical test cases.

2017) and also with recent numerical studies that consider some of the most widely used sea ice models (Mehlmann et al., 2021b).

For all combinations of velocity and dG degrees, linear kinematic features are well resolved and the deformation field is stable (for a detailed discussion, we refer to Sect. 5.3.1). The results of Fig. 9 suggest that the role of the advection scheme is minor in comparison to the discretization of the velocities. This will be discussed in Sect. 5.2.2. While the cG(1) results in the

top row of Fig. 9 are comparable to the B-grid staggerings (cG(1) is the finite element equivalent of this) given in (Mehlmann et al., 2021b, Fig. 6), the high order cG(2) results show patterns that are at least as resolved as the CD-grid results in (Mehlmann et al., 2021b, Fig. 6, Fig. 7).



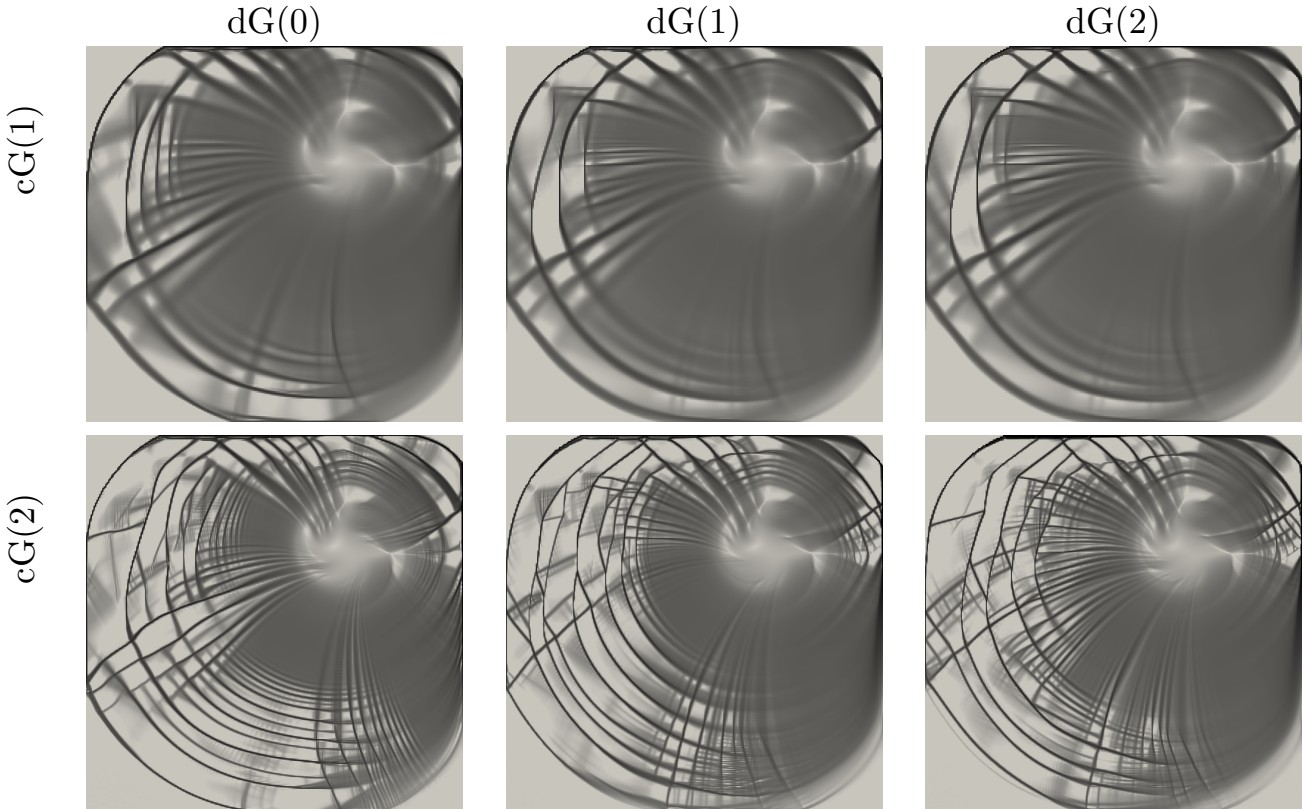

**Figure 9.** Shear deformation ($\log_{10}$) for the benchmark at time $T = 2\,\mathrm{days}$. We compare different combinations of cG velocity discretizations (linear and quadratic) with dG advection discretization (constant, linear and quadratic). All computations are run on meshes with the grid spacing $h = 2\,\mathrm{km}$.

### 5.2.1 Resolution of LKF's

To further investigate the ability of the cG/dG framework to resolve linear kinematic features, we use code provided by Hutter
et al. (2019)[1] that identify LKFs from the shear deformation rate field. The original scripts have been slightly modified in
the following manner: the resolution of the uniform quadrilateral mesh onto which the outputs are initially projected has been
increased from $256 \times 256$ to $512 \times 512$ to account for the fact that the higher order dG-discretizations carry subgrid information
that would otherwise be lost. The length of LKFs, which the scripts measure in pixels, was adjusted accordingly. Fig. 10 shows
the results for a selection of the originally published data sets (Mehlmann et al., 2021b, a) together with the results of the
low order cG(1)/dG(0) and high order cG(2)/dG(2) simulations performed with the proposed discretization. The low-order
results are consistent with the data published by Mehlmann et al. (2021b). In particular, the results agree with those obtained

---

[1]The scripts are available in the repository (Mehlmann et al., 2021a).





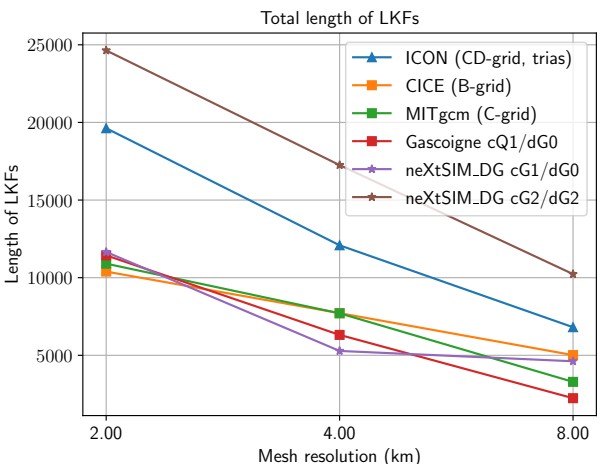
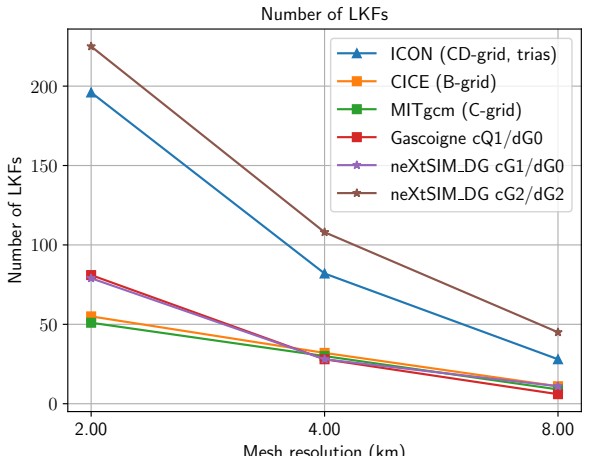

**Figure 10.** Total length and number of linear kinematic features detected for the low order $cG(1)/dG(0)$ and high order $cG(2)/dG(2)$-schemes. For comparison, we show selected simulation results on quadrilateral meshes from various models taken from Mehlmann et al. (2021b).

with Gascoigne (B-grid) (Braack et al., 2021), which is based on the same discretization. The high-order cG(2) scheme of our discretization can resolve substantially more (and longer) features on coarser meshes, demonstrating the advantage of higher order schemes.

We note that the appearance of features is strongly affected by the chosen mEVP parameters. We have not applied any fine-tuning here but use the values given in Table 1 for all meshes and all discretizations. Moreover, a direct comparison of results obtained on quadrilaterals and triangles is difficult, so we refrain from a more detailed analysis. We refer to Mehlmann et al. (2021b, Sect. 6) for an in-depth discussion as these aspects.

### 5.2.2    The role of the advection scheme

In the literature, the role of the advection scheme in the VP model and in particular for the LKF resolution is not entirely clear (again, see (Mehlmann et al., 2021b, Sect. 6)). We therefore compare for the above benchmark results for a high order momentum discretization (biquadratic velocities) and dG(0), dG(1) and dG(2) advection to shed further light on the effect of the advection.

Fig. 11 shows the ice concentration at time $T = 2\,\text{days}$ for the benchmark problem run on a mesh with spacing $h = 8\,\text{km}$.
The velocity is discretized bi-quadratically with cG(2) and the tracers are represented as discontinuous dG(0), dG(1) and dG(2) functions. The elevated surfaces in the middle and on the right in Fig. 11 shows the additional information that is gained by higher-order approaches. The finite volume dG(0) discretization only gives average values in each element while the dG(1) discretization includes the slope of the tracers and starting with dG(2) further information on the subgrid-scale, e.g. on the





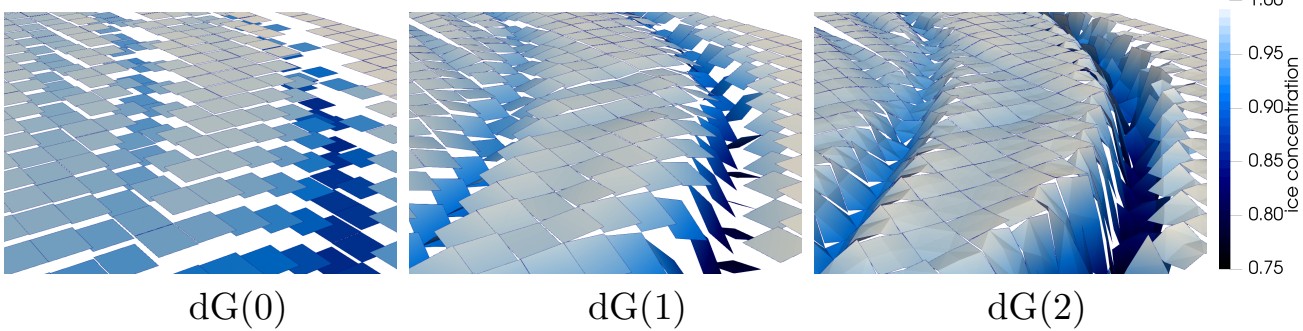

**Figure 11.** Visualization of the ice concentration on meshes with a spacing of $h = 8\,\mathrm{km}$ for dG(0) (finite volumes), dG(1) and dG(2).

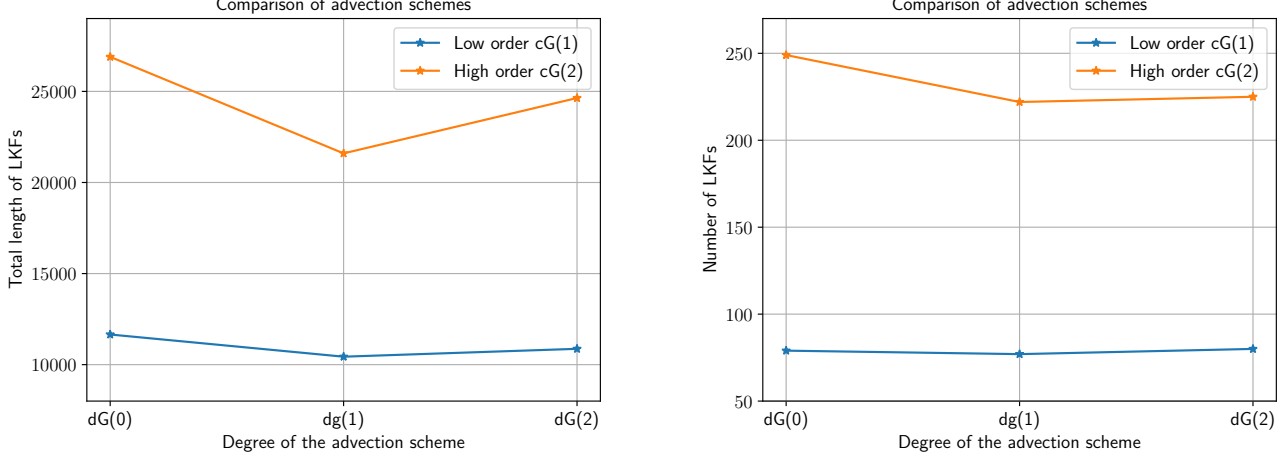

**Figure 12.** Effect of the advection scheme on resolving linear kinematic features. Left: total length of detected LKFs. Right: number of LKFs.

curvature, is also represented. The effect of the dG degree on the representation of the sea ice drift (and derived values like the shear deformation) is less drastic, as shown in Fig. 9. In comparison to the choice of the velocity discretization, the effect of the tracer discretization on the representation of LKFs is small. THis is also the conclusion of Mehlmann et al. (2021b). Fig. 12 shows the results of the LKF detection code by Hutter et al. (2019) for different advection schemes. While the low order discretization with cG(1)-velocities is hardly affected by the advection discretization, the impact on the cG(2) high-order scheme is larger. Here, the lowest order upwind scheme dG(0) yields the most and longest LKFs.



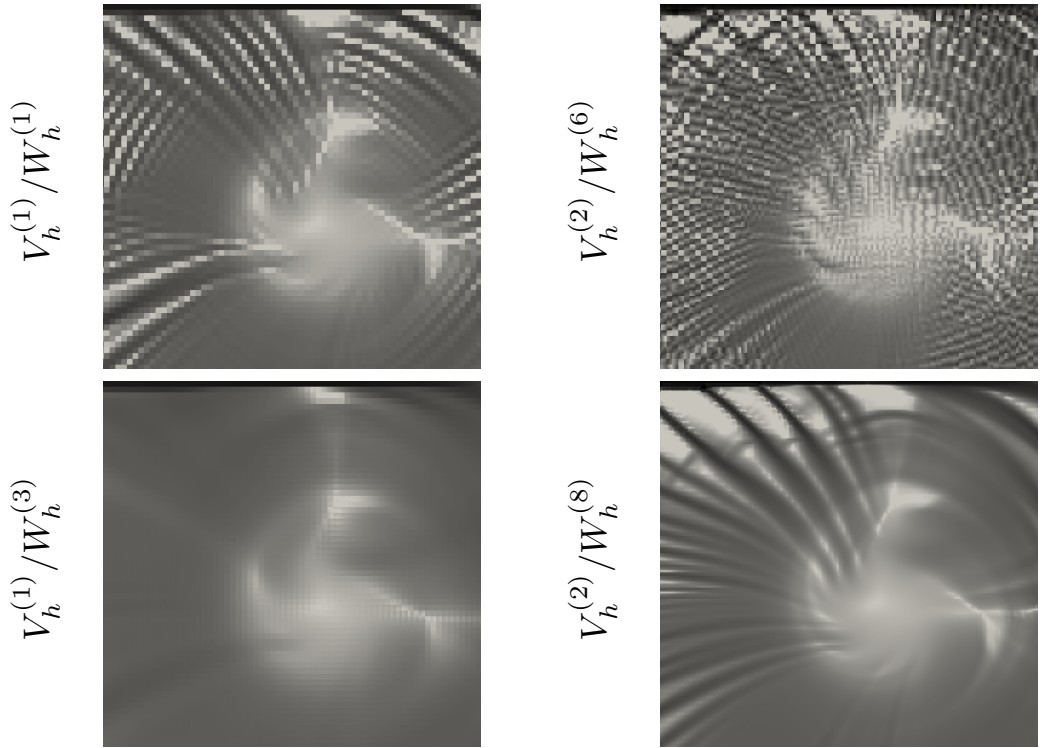

**Figure 13.** Shear at final time $T = 2$ for different choices of the velocity space $V_h^{(r)}$ and stress space $[W_h^{(s)}]^{2 \times 2, sym}$ (detail from the simulation domain). Top: unstable finite element pair with stresses one degree below the velocity space. Bottom: stable combinations.

### 5.2.3 Stability of the mixed finite element formulation

Section 3.4 introduced the mEVP iteration as a mixed finite element formulation and in particular Remark 1 discussed the optimal choice of the velocity and stress spaces. In this section, we demonstrate the effect of this choice on the results. We again consider the benchmark problem of Sec. using a mesh with $h = 2\,\mathrm{km}$. The tracers are discretized with dG(1), all parameter values are as given in Table 1.

Fig. 13 shows a snapshot of the shear deformation rate at time $T = 2\,\mathrm{days}$ for all different combinations of velocity and stress spaces. The plots clearly show the need to use large-enough stress spaces. For the low order velocity $V_h^{(1)}$ the results based on piece-wise constant stresses $\boldsymbol{\sigma} \in [W_h^{(1)}]^{2 \times 2, sym}$ (upper left) appears to give reasonable results, in particular when compared to the highly diffusive combination $V_h^{(1)}/W_h^{(3)}$ (lower left). $V_h^{(1)}/W_h^{(1)}$, however, is unstable and does not satisfy (22). It shows oscillations on the level of the mesh elements, while the combination $V_h^{(1)}/W_h^{(3)}$ is stable. Stress spaces that are too small do not provide sufficient control of the term $(\boldsymbol{\sigma}(\mathbf{v}), \boldsymbol{\Psi})$ in (19) and lead to oscillatory stresses, see the upper right plot in Fig. 13. We refer to the discussion in Section 3.4.



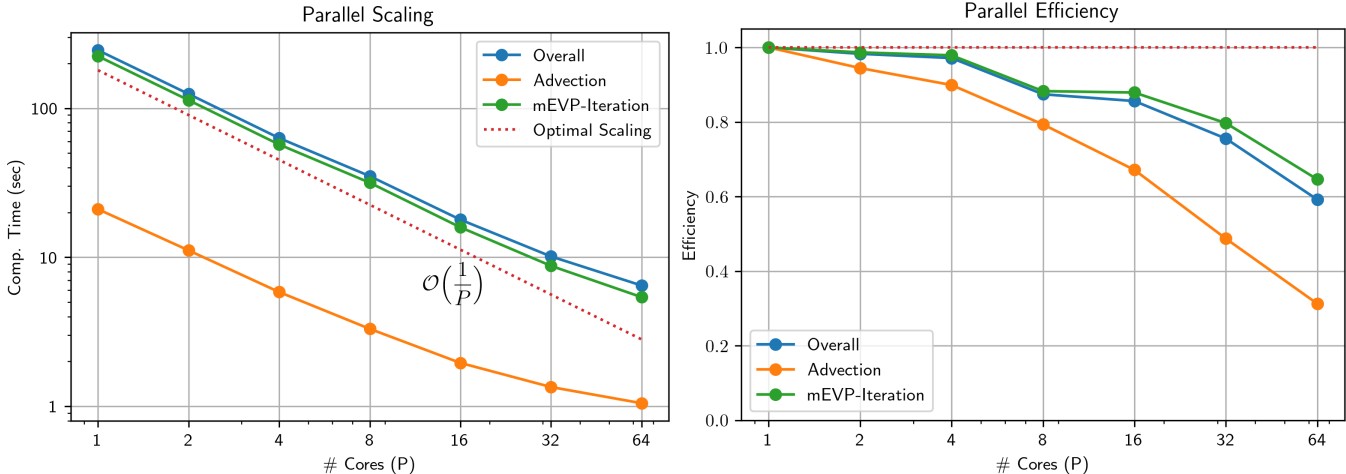

**Figure 14.** Scaling (left) and parallel efficiency (right) of the OpenMP parallelization for the benchmark test case on the $2\,\text{km}$ mesh. The mesh has $65\,536$ elements, a total of 60 advection steps with 10 momentum substeps within the mEVP iteration are covered by the total time.

## 5.3 Computational efficiency

### 5.3.1 Shared memory node-level parallelization

The complete code is parallelized using OpenMP to benefit from shared memory parallelism available on individual nodes. To
485 evaluate the parallel performance of the code, we run a strong scalability test. The benchmark problem described in Sect. is run for $T = 1\,\text{hour}$ on a mesh with $h = 2\,\text{km}$ spacing which amounts to $N = 256^2 = 65\,536$ elements. Using a time step size of $\Delta t = 60\,\text{s}$ a total of 60 advection time steps and mEVP subiterations with 100 steps each are computed. We discretize the velocity with quadratic cG(2) elements and use dG(2) with six local unknowns for ice height and sea ice concentration.

The simulation is run on an *AMD EPYC 7662 64-Core Processor* at 3.20 GHz. Fig. 14 shows the strong scalablity results.
The overall runtime drops from $245\,\text{sec}$ on 1 core to $6.5\,\text{sec}$ on 64 cores. The parallel efficiency stays very high at about $0.9$ when run on up to 16 cores and then slightly drops. The parallel efficiency of the advection scheme is not as good as the efficiency of the mEVP iteration. In this benchmark problem, this is not significant since only two tracers are advected and since there are 100 sub-steps of the mEVP solver in each advection step. For more complex thermodynamics, the situation will be different and further optimizations appear to be necessary. However, the parallelization effort so far has been restricted to
enabling OpenMP. A GPU implementaiton also provides significant opportunities for better parallel scaling.

### 5.3.2 Vectorization

Our implementation described in Sect. 4 benefits from the vectorization capabilities of *Eigen* (Guennebaud et al., 2010). *Eigen*can in particular exploit the additional computations that arise in a higher order discontinuous Galerkin methods as the





dG space $W_h^{(s)}$ naturally leads to many $s \times s$ matrices. Furthermore, the use of parametric meshes makes numerical quadrature with Gauss rules necessary and leads to operations involving $n_q \times n_{\text{cG}}$-matrices, where $n_q$ is the number of Gauss points in an element (usually 9) and $n_{\text{cG}}$ is the local number of cG basis functions (4 for cG(1) and 9 for cG(2)). The following table shows the effect of CPU-level vectorization.[2]

|  | cG(2)/dG(2) | cG(2)/dG(1) | cG(1)/dG(1) |
|---|---|---|---|
| no vectorization | 21.20 s | 18.06 s | 6.42 s |
| with vectorization | 16.43 s | 15.38 s | 4.85 s |
| acceleration | 23 % | 18 % | 24 % |

The effect is significant and purely based on the design principle in our implementation to use Eigen as much as possible. Computations are run using 16 cores of the *AMD EPYC 7662 64-Core CPU* running at 3.20 GHz and reported is the average of three consecutive calls.

### 5.3.3  Computational overhead of parametric meshes

As detailed in Sect. 2, with a parametric mesh the variational formulation needs to be mapped back onto the reference element for numerical quadrature. This has substantial computational overhead compared to fully uniform grids where all essential quantities can be precomputed. However, we described in Sect. 4 that also in the parametric case substantial precomputations are possible.

Table 2 show memory usage and computation times as a function of the mesh type and for three successively refined meshes. We also provide computational time and memory consumption per element of the mesh. We again solve the benchmark problem of Sect. 5.3.1 but only in the short interval $[0, 1\,\text{hour}]$ using 16 CPU cores and a high order discretization, i.e. cG(2) with dG(2) advection. The results in Table 2 clearly show the superiority of the uniform mesh approach both in terms of computational time and memory consumption. It shows, however that parametric meshes can be realised either with a comparable memory footprint but with substantial computational overhead or with comparable computational efficiency but with increased memory requirements.

The trade-off that is chosen in an implementation will likely depend on the hardware that is targeted. On common multi-core CPUs with a moderate number of cores, the parametric approach with precomputed matrices seems to be superior. When using many-core systems or moving to GPUs or TPUs, the balance between compute-memory performance may differ. This remains a task for further research in the future.

---

[2] Vectorization can be activated or deactivated by a simple compiler flag, for instance `-march=native`, which turns on *all* hardware-specific optimization and in particular vectorization support.





| mesh size $N_{el}$ | 4096 | | 16384 | | 65536 | |
|---|---|---|---|---|---|---|
| memory usage | 11.8 MB | 2.88 kB/N$_{el}$ | 26.2 MB | 1.64 kB/N$_{el}$ | 81.9 MB | 1.25 kB/N$_{el}$ |
| comp. time | 5.77 s | 1.41 ms/N$_{el}$ | 27.11 s | 1.65 ms/N$_{el}$ | 109.96 s | 1.67 ms/N$_{el}$ |

(a) Parametric meshes, on-the-fly computation of the mapping.

| mesh size $N_{el}$ | 4096 | | 16384 | | 65536 | |
|---|---|---|---|---|---|---|
| memory usage | 21.4 MB | 5.22 kB/N$_{el}$ | 70.2 MB | 4.28 kB/N$_{el}$ | 259 MB | 3.95 kB/N$_{el}$ |
| comp. time | 1.40 s | 0.35 ms/N$_{el}$ | 5.76 s | 0.35 ms/N$_{el}$ | 22.47 s | 0.34 ms/N$_{el}$ |

(b) Parametric meshes, precomputed matrices for fast mapping.

| mesh size $N_{el}$ | 4096 | | 16384 | | 65536 | |
|---|---|---|---|---|---|---|
| memory usage | 11.4 MB | 2.78 kB/N$_{el}$ | 24.5 MB | 1.50 kB/N$_{el}$ | 74.3 MB | 1.13 kB/N$_{el}$ |
| comp. time | 1.48 s | 0.36 ms/N$_{el}$ | 4.44 s | 0.27 ms/N$_{el}$ | 15.69 s | 0.24 ms/N$_{el}$ |

(c) Uniform meshes, precomputed static matrices.

**Table 2.** Comparison of different mesh structures. Evaluation of the benchmark problem with $cG(2)/dG(2)$ discretization on meshes of size $h = 8\,\text{km}$, $h = 4\,\text{km}$ and $h = 2\,\text{km}$. The benchmark is simulated for $T = 1\,\text{hour}$. Tables 2a and 2b refer to the parametric meshes, where the variational formulation must be evaluated using Gauss quadrature, 2c corresponds to the uniform mesh, where the exact evaluation of the variational form is hard-coded. 2b in comparison to 2a precomputes and stores the matrices required for the parametric mapping.

## 6 Conclusion

We presented the numerics and implementation of the neXtSIM-DG dynamical core, a new discretization of sea ice dynamics aimed at Earth system models. A key feature is the use of higher order in terms of local discretization and the consistent use of efficient data structures and modern programming paradigms.

The new framework has been validated in the context of Hibler's established viscous-plastic sea ice model but the dynamical core is flexible and can accommodate different rheology models.

All advection equations are discretized using discontinuous Galerkin methods. Currently, the methods dG(0), dG(1), and dG(2) have been implemented and validated, but the flexible software concept based on code-generation and pre-assembled matrices for efficient implementation of the variational formulations easily allows an extension to even higher orders. This could become relevant in connection with alternative sea ice rheologies, see e.g. Dansereau et al. (2016); Rampal et al. (2016). The momentum equation is discretized using second order continuous finite elements.



We validated the advection discretization and showed that the theoretically expected orders of convergence are realized
in practice. Thereby, by using parametric grids, we achieve great flexibility on the spatial discretization. On the other hand,
the underlying structured grid topology allows for an efficient numerical implementation. The momentum equation with an
mEVP approximation of the visco-plastic model and was tested using an established benchmark problem Mehlmann and
Richter (2017). In particular, we showed that the high-order discretization can resolve more LKFs than the established models Mehlmann et al. (2021b), albeit with a larger number of degrees of freedom.

We also described the implementation of the dynamical core on a shared-memory compute node and parallelization using
OpenMP. In our future work, we will add coarser parallelization on distributed clusters with MPI and also parallelization on
GPUs.

While MPI parallelization is standard and can be easily accomplished by using topologically simple, structured rectangular
grids, we enter new territory with GPU parallelization. However, this is already prepared by using a structured memory design.
Moreover, the current implementation of neXtSIM-DG allows an easy choice between a pre-assembly of matrices and an
on-the-fly computation of all sizes, which could be beneficial on GPUs to reduce memory bandwidth, see Section 5.3.3.

## 7 Code availability

The software project neXtSIM-DG is under active development and hosted on GitHub, https://github.com/nextsimdg. A snapshot including the scripts to reproduce the examples of this manuscript is published as Zenodo repository (Richter et al., 2023).

## 8 Author contribution

All authors have worked on the concepts for the design of neXtSIM-DG. TR, CL and PM designed and implemented the
software framework. All authors worked on the draft of the paper, and jointly submitted to the validation, discussion and
presentation of the results as well as the final editing of the manuscript.

## 9 Competing interests

The authors declare that they have no conflict of interest.

## 10 Acknowledgements

All authors are supported by the project SASIP (grant no. 353) funded by Schmidt Futures – a philanthropic initiative that
seeks to improve societal outcomes through the development of emerging science and technologies.



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
