# Peer review of "The neXtSIM-DG dynamical core: A Framework for Higher-order Finite Element Sea Ice Modeling"

_EGUsphere, 2023_

## Author Response (AR1)

**Review 1**

It is a well written paper documenting a finite-element based approach to modeling sea ice dynamics in the framework of viscous plastic rheology. The approach is based on quadrilateral elements and explores both bilinear and bi-quadratic representation for sea ice velocity. The tracers are represented using discontinuous elements, but the authors find that the sensitivity to the accuracy of tracer advection is not very high. I recommend publishing this manuscript after minor revision.

I would recommend to briefly discuss the case of nonconforming (CR) element. According to earlier studies, it provides a very high resolution, and will similarly perform very close to the bi-quadratic case. The computational load is in this case seemingly lower.

*We added a comment on nonconforming elements in the introduction. Furthermore, for our test case described in Section 5.2.1 we already provide a comparison to these. The results are, indeed, similar when the number of LKFs is considered. Furthermore, when the total length is taken as measure then biquadratic elements provide performance comprable to linear elements with one additional refinement level.*

Also, I would suggest to add numbers allowing one to judge about the computational time required by bilinear and bi-quadratic cases. The first one largely corresponds to the approach taken by CICE, so the numbers will be helpful to judge on how the common type discretization is related to the higher-order one.

*We have added a table with computational times comparing the different approaches. It is Table II with a discussion in Section 5.2.1.*

**Minor points:**

- line 14 'It is, for example, an important part ....' – It contributes importantly to the global ...
  *corrected*

- 15 circulation
  *corrected*

- 62 'to limit diffusion'? Incremental remapping helps to ensure positivity. It relies on limiting, so it is not clear to what an extent it is limiting diffusion.
  *You are right. We changed it to "to preserve monotonicity"*

- 83 It is the mean height (volume per unit area)
  *corrected*

- 117 remove b
  *corrected*

- 120 It is worthwhile to mention that this is never achieved in practice and mEVP commonly deals with non-converged solutions.
  *We have added a note and a citation regarding the use of mEVP.*

- Section 3.1. The discussion of limitations due to advection is not really relevant. First, $\Delta t$ is generally governed by the ocean model, where velocities are larger, so sea ice CFL will not be an issue. Second, there are internal stresses that depend on the mean thickness and concentration, and this may lead to a limitation
  *We do not fully agree as the CFL condition scales with the degree of the DG approach like $(2r+1)^{-1}$. In our benchmark configurations we, indeed, got very close to the CFL limitation on very fine meshes.*

- 130 Sea ice was sometimes run with larger time step than ocean, but I am not aware about the situation described here.
  *As this discussion is not relevant for our work we removed it from the text.*

- 133 Why this range? It will obviously depend on mesh resolution.
  *You are right. We revised the text and now state that it i just one example (and not necessarily "typical").*

- 135 This is the limitations due to advection, it may matter for high-order in principle, but as I wrote, it is commonly ocean that matters most, and there is a wave-type limitation due to plastic response.
  *Thanks. We again stressed the fact that this discussion is relevant due to the high order advection only.*

- 140 To be consistent ... – MPAS, ICON and FESOM communities occupy a substantial part of climate modeling, and they rely on different meshes
  *We have removed this sentence. Quadrilateral meshes are indeed just one choice among others.*

- 141 unstructured –¿ distorted
  *corrected*

- 144 for a depiction –¿ for an illustration
  *corrected*

- Expressions (12) and (13): the lower indices start from 0 in (13), and 1 in (12)
  *corrected*

- 186 Which order of RK is used?
  *We started to take the same order for time-stepping as for the spatial DG scheme, so explicit Euler and second and third order RK. At the moment we think that using a second-order RK method is a good and efficient choice. The results in Sec. 5 of our paper show that advection does not play a dominant roles.*

- 212 The Babushka-Brezzi condition is a subject of many publications and is well know, so at least include (see, e.g. Ern ...)
  *added.*

- Remark 1 – Why 'Optimality...'? Say before that the discretization on quads involves a lot of DoFs and computations, and a question arises on its optimality with respect to triangular meshes where functional spaces seem smaller.
  *We changed the title of this remark to "Mixed velocity-stress discretization on triangular and quadrilateral meshes". We also rephrased the remark to make the statement clearer: quadrilaterals require a larger stress space. But considered globally, the difference is small compared to a triangular mesh.*

- 256 by Gaussian
  *corrected*

- 325 a triangle – an element
  *corrected*

- Table 1 Ice concentration?
  *We now write "ice concentration parameter"*

- 450 is strongly affected? I do not think it is the case. As soon as stability is ensured, sensitivity is generally very moderate.
  *We have toned this down and now only state "number and length of features are affected ..."*

- 466 THis
  *corrected*

- 5.2.3. I do not think this should go in the main text, move it to an appendix
  *We would like to keep the section in the main part of the paper to not interrupt the reading flow; the section would also be very isolated as the only appendix. The demonstration of the need for a sufficiently large stress space is also very important in our opinion.*

- Figure 14, caption '10 momentum steps' — 10 or 100?
  *corrected*

- 5.3.1. The efficiency on shared memory level is perhaps not surprising given the measures described. The real challenge, however, is the MPI parallelism, because of the large number of substeps in mEVP.
  *Finite element simulations are almost always limited by memory bandwidth. In this respect, they are highly unfavorable for modern compute hardware since only few computations are incurred per degree of freedom. Therefore it is often very difficult to achieve good shared-memory scalability. The usual way to achieve good parallel performance is a matrix-free approach and the use of higher order elements. This can be seen in Table 2 (new version), which shows very good scaling from cG1 to cG2, far better than can actually be expected. On the other hand, the calculations in 5.3.3 show that for cG2 the point has not yet been reached where it is clearly advantageous to dispense with precalculated matrices. All in all, shared-memory parallelization of FE methods is still not a completely solved problem. In this*

*respect, MPI scaling is easier, since here the memory bandwidth grows parallel to the number of nodes.*

**Review 2**

The authors present a finite-element model of sea-ice under elastic-visco-plastic approximation. Specifically, they chose a dG advection scheme for sea-ice concentration, and a mixed approach for the momentum equation, with continuous Galerkin method for all terms except the stress tensor, which was discretized with dG method. In the numerical tests, the authors focus on how different order cG and dG discretizations affect the sea-ice shear deformation, and study computational performance of their code parallelized using shared memory paradigm. Despite the increased computational cost, higher order discretization seem to provide superior results and much smaller diffusion. I am glad that authors consider moving to even higher basis polynomial orders, as the increased computational per-element intensity can lead to better parallel scalability results, offsetting the increased cost.

The paper is well written and contains sufficient math background, numerical tests and physics validation to merit publication. I do not have recommendation for revisions.

*Many thanks for this positive review.*

---

## Author Response (AR2)

Dear editorial team,

We've uploaded the final version of our manuscript including all source files. We have removed all highlighting marking the changes of the revised version. Otherwise, we have not applied any changes.

The email announcing the decision has indicated the necessity of some "technical changes" and referred to the editor's report. We have however not found any further comments.

If we missed something we'll of course quickly edit the files.

Best regards,

Thomas Richter